# ZipLM: Inference-Aware Structured Pruning of Language Models

**Eldar Kurtic**
IST Austria
eldar.kurtic@ist.ac.at

**Elias Frantar**
IST Austria
elias.frantar@ist.ac.at

**Dan Alistarh**
IST Austria & Neural Magic
dan.alistarh@ist.ac.at

## Abstract

The breakthrough performance of large language models (LLMs) comes with major computational footprints and high deployment costs. In this paper, we progress towards resolving this problem by proposing a novel structured compression approach for LLMs, called ZipLM. ZipLM achieves state-of-the-art accuracy-vs-speedup, while matching a set of desired target runtime speedups in any given inference environment. Specifically, given a model, a dataset, an inference environment, as well as a set of speedup targets, ZipLM iteratively identifies and removes components with the worst loss-runtime trade-off. Unlike prior methods that specialize in either the *post-training/one-shot* or the *gradual compression* setting, and only for specific families of models such as BERT (*encoder*) or GPT (*decoder*), ZipLM produces state-of-the-art compressed models across all these settings. Furthermore, ZipLM achieves superior results for a fraction of the computational cost relative to prior distillation and pruning techniques, making it a cost-effective approach for generating an entire family of smaller, faster, and highly accurate models, guaranteed to meet the desired inference specifications. In particular, ZipLM outperforms all prior BERT$_{base}$ distillation and pruning techniques, such as CoFi, MiniLM, and TinyBERT. Moreover, it matches the performance of the heavily optimized MobileBERT model, obtained via extensive architecture search, by simply pruning the baseline BERT$_{large}$ model. When compressing GPT2, ZipLM outperforms DistilGPT2 while being 60% smaller and 30% faster. Our code is available at: https://github.com/IST-DASLab/ZipLM.

## 1 Introduction

The high accuracy of modern language models from the Transformer family [53] comes at the price of massive computational cost, which hinders their practical adoption in resource-constrained settings. This has motivated the development of *model compression* techniques, which can be categorized into *pruning* [17], *quantization* [9], and *distillation* [11]. In this paper, we focus on *structural compression*, whose goal is to reduce model size by removing entire sub-components, such as rows or columns from the model's weight matrices. The key advantage of structured pruning, relative to unstructured pruning of individual weights, is that the model can be reshaped to new dimensions, and the resulting computational savings can be leveraged on any hardware, without specialized computational support. At the same time, structured pruning introduces significant challenges. First, models are usually highly-sensitive to structured compression, and most methods require *gradual compression*, including retraining cycles designed to allow the model to recover accuracy. In addition, structural compression significantly complicates the use of knowledge distillation [16], which is usually done via manual or dynamic layer mapping [21, 59]. On the practical side, another challenge is that most existing techniques do not provide *runtime speedup* guarantees: the model is pruned to a fixed sparsity or FLOPS target, and then must be evaluated in the target inference environment. If the pruned model fails to meet the target inference specifications, the whole process must be repeated from scratch.

37th Conference on Neural Information Processing Systems (NeurIPS 2023).

**Overview.** In this paper, we resolve these issues and provide a novel structured pruning approach called ZipLM, which achieves state-of-the-art performance, both in the *post-training/one-shot* setting, where retraining is not desirable, as well as in the popular *gradual compression* setting, where retraining is possible. We accomplish this via an inference-aware algorithm, which successfully balances the loss-runtime trade-off at each pruning step. By taking runtime into account, we avoid removing components that do not bring significant speedup gains. Additionally, our algorithm provides speedup guarantees for compressed models, a highly-desirable property in practical applications.

We summarize our contributions as follows:

- We introduce a novel structured pruning approach, which unifies the saliency criteria investigated by prior work–weight magnitude, activation impact, and removal of linearly-redundant structures, while considering local (layer-wise) and global correlations. We augment it to be *inference-aware*, ensuring desired latency or throughput in any given configuration.

- We complement the algorithm with a novel *layer-wise token-level distillation*, which consistently boosts accuracy on small datasets and does not require manual layer matching, circumventing a limitation of prior structured pruning techniques.

- ZipLM is the first structured pruning approach that achieves state-of-the-art results for both, *post-training/one-shot* compression and *gradual pruning* settings, while being applicable to both, BERT (*encoder*) and GPT (*decoder*) language models, without any modifications.

- ZipLM is practical and efficient. For a set of desired speedups (e.g. 2x, 5x, 10x) in the target inference environment (e.g. batch-size=128, sequence-length=384, device=V100), in a single run and under the same set of hyper-parameters, it produces the entire family of compressed models, one for each speedup target. Consequently, it leads to state-of-the-art results in *GPU-based* inference environments. Moreover, it is compatible with unstructured pruning and quantization, leading to state-of-the-art results even for *CPU-based* environments.

## 2   Related Work

**Distillation-based compression methods** focus on training a smaller student model to mimic the representations of a larger teacher model. The "distance" between the representations of student and teacher is often architecture-specific. MiniLM [55] uses a deep self-attention mechanism to replicate the attention mechanism of the teacher, and TinyBERT [21] employs a bespoke distillation mechanism, for a manually-picked subset of layers. Both methods offer a very strong baseline, generally outperforming other approaches, except for MobileBERT. MobileBERT [51] involves first training a custom large BERT teacher model from scratch, and then deviates from the standard architecture [2] by introducing heavily-optimized components with reduced latency, whose combinations are decided in neural architecture search (NAS)-like fashion. It achieves strong results in terms of accuracy-per-parameter, at the cost of significant computational costs in the search process. DistilBERT and DistilGPT2 [44] involve training a fixed student obtained by removing every other layer from the teacher, while BERT-PKD [50] employs incremental knowledge extraction through the distillation of intermediate layers. Well-Read-Students [52] reduces the size of the standard BERT architecture through principled downscaling of internal dimensions. DynaBERT [18], on the other hand, distills knowledge to a student model that is both depth- and width-adaptive.

**Structural pruning methods** usually start from a large pre-trained model, and iteratively reduce the dimensions of weight matrices. Block Movement Pruning [27] identifies and removes redundant rectangular blocks of weights while following the movement pruning intuition [45] that weights moving towards zero during fine-tuning should be removed. FLOP [56] and Low-Rank [40] use matrix decomposition techniques to progressively remove rank-1 components from factorized weight matrices during training. BERT-of-Theseus [60] employs a similar approach, but replaces entire submodules with smaller counterparts. Methods like LayerDrop [3] and Poor Man's BERT [43] address structured compression through various layer-dropping techniques. LayerDrop uses structured layer-dropout regularization to train a model resilient to sub-network selection during inference, while Poor Man's BERT explores a wide range of layer-dropping strategies. The recent CoFi method [59] employs masks of different granularities to jointly prune coarse and fine-grained submodules during fine-tuning, combined with an optional customized distillation technique. CoFi is the state-of-the-art *structural pruning* method; relative to distillation methods, CoFi outperforms MiniLM and TinyBERT, but not MobileBERT, in terms of accuracy-vs-speedup.

**Other compression methods** such as the ones that exploit dynamic forms of sparsity which appear at runtime [35], or the ones that utilize lower bit-width representation of weights and/or activations [6, 32] are complementary to our approach. We demonstrate this in Section 5 where we apply quantization to obtain even higher compression ratios for edge deployment environments like commodity CPUs.

## 3 Method

Removing large structures like entire matrix columns or attention heads from a language model quickly leads to severe accuracy degradation, from which it is often difficult to recover even with extensive finetuning. This is why current state-of-the-art approaches like Block Movement Pruning [27] or CoFi [59] opt for integrating pruning directly into training (via sampling or differentiable approximations), rather than performing it in the standard gradual pruning fashion of discrete steps with finetuning in between. However, as we will show, by designing a new highly accurate pruning algorithm which is able to account for both local correlations of structures within single layers as well as global correlations across layers, we can actually apply the gradual pruning paradigm, with all its advantages, to improve significantly over the current state-of-the-art.

### 3.1 The ZipLM Structured Pruning Algorithm (Local Correlations)

Most existing structured pruning criteria [33, 31] are based on one or two of the following assumptions about saliency: structures with lower (average) weight magnitude are easier to prune [15, 29], structures with small input activations can be removed at little loss [34], and structures that are close to a linear combination of other structures are the most redundant [14, 49]. We will now show how all these aspects can be jointly considered in a principled manner via our new ZipLM technique.

**Problem formulation.** Our approach starts from the idea of applying structured compression layer-wise, in a way that allows the layer to preserve most of its output characteristics. This setup is popular in the post-training quantization and unstructured pruning literature [38, 19, 8], and can be implemented as follows. We are given a small amount of calibration data, which we run through the network, to obtain "reference" inputs and outputs for each layer. Then, for each layer, given the calibration inputs $\mathbf{X}$ and the original layer weights $\mathbf{W}$, we aim to find compressed weights $\widehat{\mathbf{W}}$ respecting the compression constraint $\mathcal{C}$, which best approximate the original output, measured via the squared error metric. If we assume that the input and weight matrices have an appropriate rectangular form, the problem can be formalized as:

$$\text{argmin}_{\widehat{\mathbf{W}}} \, ||\widehat{\mathbf{W}}\mathbf{X} - \mathbf{W}\mathbf{X}||_2^2 \quad \text{subject to} \quad \widehat{\mathbf{W}} \in \mathcal{C}. \tag{1}$$

This objective can be decomposed across the rows of $\mathbf{W}$, leading to a set of sparse linear regression problems, one per row. These row-wise problems are independent, which forms the basis of related work [8]; yet, since we do *structured* pruning, they become dependent, as we would like to prune the same weight indices *across all rows*, i.e. prune entire columns. Thus, finding the optimal weights $\widehat{\mathbf{W}} \in \mathcal{C}$ is equivalent to finding: 1) the optimal structure $\mathbf{S}$ of the desired shape to be removed, which we assume to be applied across all rows, with corresponding pruning mask $\mathbf{M_S}$, where pruned indices have value 1 in the mask, and others are 0; and 2) the corresponding update $\boldsymbol{\delta_S}$ to all of the remaining weights, optimally compensating for the error caused by the removal of weights in $\mathbf{S}$.

**Saliency scores and weight update.** Let $\mathbf{H} = \mathbf{X}\mathbf{X}^\top$ be the Hessian matrix for the $\ell_2$-minimization problem in Equation 1, which is independent of the weights. Define $\mathbf{W}_{i,\mathbf{M_S}}$ to be the subset of weights under the mask $\mathbf{M_S}$ in row $i$, and by $(\mathbf{H}^{-1})_{\mathbf{M_S},\mathbf{M_S}}$ the submatrix of the inverse Hessian corresponding to the entries under the mask $\mathbf{M_S}$. Then, we can obtain the optimal mask and weight update as follows:

$$\text{argmin}_{\mathbf{S}} \sum_{i=0}^{d_{\text{row}}} \mathbf{W}_{i,\mathbf{M_S}} \cdot \left( \left(\mathbf{H}^{-1}\right)_{\mathbf{M_S},\mathbf{M_S}} \right)^{-1} \cdot \mathbf{W}_{i,\mathbf{M_S}}^\top \tag{2}$$

$$\boldsymbol{\delta_S} = -\mathbf{W}_{:,\mathbf{M_S}} \cdot \left( \left(\mathbf{H}^{-1}\right)_{\mathbf{M_S},\mathbf{M_S}} \right)^{-1} \cdot \left(\mathbf{H}^{-1}\right)_{\mathbf{M_S},:} \tag{3}$$

We obtain this by extending the Optimal Brain Surgeon [13, 24] formulas for solving Equation 1 to cover all $d_{\text{row}}$ weight matrix rows simultaneously. Importantly, the subselection of the inverse

Hessian $((\mathbf{H}^{-1})_{\mathbf{M_S},\mathbf{M_S}})^{-1}$ is shared between all rows. Further, since we generally consider only non-overlapping sets $\mathbf{S}$ of the same size, we pay just $O(d_{\text{col}} \cdot |\mathbf{M_S}|^2)$ total cost for all extra inversions. Since the number of structures in the mask $|\mathbf{M_S}|$ is usually small, e.g. attention heads usually consist of 64 columns, the overall cost of these inversions is low.

Simply selecting the structures to prune according to the criterion in Equation 2 unifies the weight magnitude and activation influence criteria (via the Hessian), but still ignores any correlations between structures. We address this by pruning structures *one-at-a-time*, while always applying update $\boldsymbol{\delta_S}$ and fully recomputing $\mathbf{H}^{-1}$ relative to the remaining structures. For example, if there exist two redundant structures $S_1$ and $S_2$, we will first drop $S_1$ and update $S_2$ to compensate for this removal, at which point $S_2$ is no longer easy to prune. Without this one-at-a-time removal, both structures would have been incorrectly removed as they each individually seem easy to prune according to Equation 2. Executing this strategy naively will require a full $O(d_{\text{col}}^3)$ recomputation of the inverse Hessian relative to the remaining structures at each step, which would be very slow. However, this can be avoided by removing the rows and columns corresponding to $\mathbf{M_S}$ directly in the inverse with one step of Gaussian elimination [8], applied block-wise to cover larger structures, as follows:

$$\mathbf{H}^{-1} - \mathbf{H}^{-1}_{:,\mathbf{M_S}} \cdot \left((\mathbf{H}^{-1})_{\mathbf{M_S},\mathbf{M_S}}\right)^{-1} \cdot \mathbf{H}^{-1}_{\mathbf{M_S},:}, \tag{4}$$

which takes only $O(|\mathbf{M_S}| \cdot d_{\text{col}}^2)$ time. We provide complete pseudocode in Algorithm 1.

---

**Algorithm 1** The ZipLM pruning algorithm. Given inverse Hessian $\mathbf{H}^{-1} = (2\mathbf{X}\mathbf{X}^\top + \lambda\mathbf{I})^{-1}$, we remove exactly $k$ structures from the corresponding weight matrix $\mathbf{W}$.

---

$\mathbf{R} \leftarrow$ set of all possible structures
**for** $k$ times **do**
$\quad \mathbf{S} \leftarrow \text{argmin}_{\mathbf{S}} \sum_{i=0}^{d_{\text{row}}} \mathbf{W}_{i,\mathbf{M_S}} \cdot ((\mathbf{H}^{-1})_{\mathbf{M_S},\mathbf{M_S}})^{-1} \cdot \mathbf{W}_{i,\mathbf{M_S}}^\top$
$\quad \boldsymbol{\delta_S} \leftarrow -\mathbf{W}_{:,\mathbf{M_S}} \cdot ((\mathbf{H}^{-1})_{\mathbf{M_S},\mathbf{M_S}})^{-1} \cdot (\mathbf{H}^{-1})_{\mathbf{M_S},:}$
$\quad \mathbf{W} \leftarrow \mathbf{W} + \boldsymbol{\delta_S}$
$\quad \mathbf{H}^{-1} \leftarrow \mathbf{H}^{-1} - \mathbf{H}^{-1}_{:,\mathbf{M_S}} \cdot ((\mathbf{H}^{-1})_{\mathbf{M_S},\mathbf{M_S}})^{-1} \cdot \mathbf{H}^{-1}_{\mathbf{M_S},:}$
$\quad \mathbf{R} \leftarrow \mathbf{R} - \{\mathbf{S}\}$
**end for**
$\mathbf{W} \leftarrow \mathbf{W} \odot \mathbf{M_R}$

---

We utilize the fact that the values corresponding to pruned weights in $\mathbf{W}$ and in the inverse Hessian $\mathbf{H}^{-1}$ do not affect any subsequent calculations and can therefore be ignored even if they are not exactly zero. However, in the end we have to prune them explicitly again by multiplying with the overall mask to ensure that they are exactly zero. In a practical implementation, $((\mathbf{H}^{-1})_{\mathbf{M_S},\mathbf{M_S}})^{-1}$ should only be computed once and reused when computing the corresponding sum across all rows.

**Pruned structures.** Focusing on Transformers, we consider three types of structural removal: dropping attention heads, shrinking the expanded intermediate dimension of the fully-connected network (FC) layers, and removing entire residual parts, i.e. attention or FC-modules. We implement this by dropping $d_{\text{head}}$ consecutive columns in the out-matrix of the attention block and individual columns in the second linear layer of the feed-forward network. Once these column-structures are zeroed out, corresponding rows in previous layers can be safely removed without any output change. Crucially, by pruning e.g. columns in the FC2 layer rather than equivalent rows in FC1, we can utilize the input correlations via Hessian-information using the ZipLM pruner.

**Novelty relative to existing Optimal Brain Surgeon (OBS) approaches.** The original framework [13], as well as modern efficient versions [47, 7, 8], have been explicitly developed for *unstructured pruning*, i.e. removing individual weights. It is nontrivial to extend them to structured pruning, as this involves considering additional correlations, both within as well as across multiple blocks (such blocks are usually employed for computational tractability). For example, the state-of-the-art layer-wise approach of [8], performs unstructured pruning by handling weight matrix rows separately, and then greedily merging results. In contrast, we perform structured pruning *jointly* across multiple rows, which is not only necessary for correctness but additionally enables us to design an algorithm with a computational complexity that is lower by a full factor of the hidden dimension size. Additionally, structured pruning requires explicitly matching matrix shapes for consecutive layers and a dedicated strategy for utilizing weight updates even when entire blocks/rows are pruned.

### 3.2 Inference-Aware Structured Pruning (Global Correlations)

We now describe how to augment the algorithm to be *inference-aware*, in the sense that it accepts inference specifications, such as batch-size, sequence-length, and speedup on the target hardware, as additional inputs to optimize for.

**Motivation.** The main benefit of inference-aware structured pruning is the fact that pruning decisions are not guided purely by saliency scores, but instead by loss-vs-speedup trade-offs associated with the removal of each component in the model. Prior methods, e.g. [24, 27, 59] focus solely on pruning until a specific sparsity threshold is reached, without taking into account the real-world speedups corresponding to the compression threshold, which can vary significantly between settings. For example, a 95% sparse BERT produced by CoFi [59] has 12x speedup on a V100 GPU, but only 5x on an A100 GPU. With existing methods, if real-world timings fail to meet the inference requirements, the entire process has to be repeated with different sparsity values until the target speedup is achieved, which is both time-consuming and error-prone. An additional advantage of inference-awareness, which we showcase in our GPT experiments in Section 4, is that it enables optimizing for different real-world metrics, such as latency or throughput.

**Runtime awareness.** We integrate runtime constraints via a latency table [1] for our target inference environment, where we record the time to run an attention block, including all overheads, with $0, \ldots, N_{\text{heads}} - 1$ heads pruned and similarly for the fully-connected block with the intermediate dimension shrunk by a factor of $0.9^i$, for $i = 0, \ldots, 42$; in relative steps of 10% up until $\approx 99\%$ sparsity, following [5]. This allows rapid runtime estimation for different *per-layer sparsity configurations*. We provide an example of our latency table in Appendix E.

**Finding the optimal sparsity configuration.** Ultimately, our goal is to find a per-layer-sparsity configuration that satisfies a certain speedup-constraint while maximizing accuracy. A popular paradigm of doing this [15, 30] is to produce a large number of pruned models with different sparsity distributions across layers and then select the one, satisfying a target constraint, with the highest accuracy. To make this computationally feasible, it is crucial that pruning is cheap, yet accurate. ZipLM treats each layer independently, which makes it possible to precompute a database of several pruned versions with different sparsities for each layer. The entire database can be produced in a single run, utilizing the algorithm's one-at-a-time nature. While our algorithm is compatible with various search methods for finding layer-wise profiles [15, 12], we adapt the recent SPDY approach [5].

**Structured SPDY search.** The SPDY approach is designed for unstructured pruning and assigns a quadratic prior to per-layer sensitivity of different sparsity levels. This is not valid in our structured pruning scenario, since for instance it would suggest that dropping a full layer is only slightly more difficult than pruning it to 99% sparsity. Thus, using standard SPDY would lead the algorithm to explore a large number of sub-optimal configurations, significantly wasting computational resources. To alleviate this problem, for a structured sparsity $s$, we introduce a better prior $p_s$ as the relative layer-wise squared error incurred by pruning, defined as $p_s = ||\widehat{\mathbf{W}}_{\mathbf{s}}\mathbf{X} - \mathbf{W}\mathbf{X}||_2 / ||\mathbf{W}\mathbf{X}||_2$, which simply has a value of 1 for a fully dropped layer. Furthermore, the original SPDY approach uses shrinking neighborhood search, which has high variance in both runtime and solution quality for structured compression. Therefore, we perform a fixed number of 1000 steps, randomly mutating *in expectation* 10% of the layer-wise sensitivity coefficients. Finally, we note that any candidate evaluated by this procedure actually achieves the target speedup, leading to significantly decreased search time. We validate our approach in Appendix F, where we demonstrate that our speedup estimations are indeed very accurate in practice. Specifically, real-world on-device measurements deviate at most by 5.28% from their expected values.

### 3.3 Layer-wise Token Distillation

For structured pruning, it is common to apply *layer-wise distillation* objectives to transfer intermediate representations. However, structured pruning creates compatibility issues relative to the fixed teacher architecture, leading most methods to develop customized distillation strategies. A popular approach, introduced in [21] and improved by [59], solves the problem via static [21] or dynamic [59] mapping of a subset of teacher layers to a subset of student layers. Their main limitation is manual layer selection, where making the "optimal" choice would require evaluating all possible combinations, which can be very expensive. Another limitation is shape-matching between intermediate layers, which is solved by introducing a learnable linear transformation matrix attached to student outputs.

**Our approach.** We address these challenges differently, by leveraging the fact that ZipLM preserves the hidden dimension size, and propose to use distillation of intermediate token representations across the entire model. The resulting minimization objective consists of three components:

$$\mathcal{L}(\theta^{\text{s}}, \theta^{\text{t}}|x) = \lambda_1 \mathcal{L}_{\text{task}}(\theta^{\text{s}}|x) + \lambda_2 \mathcal{L}_{\text{logit}}(\theta^{\text{s}}, \theta^{\text{t}}|x) + \lambda_3 \mathcal{L}_{\text{token}}(\theta^{\text{s}}, \theta^{\text{t}}|x), \tag{5}$$

where $\theta^s$ and $\theta^t$ represent student and teacher models respectively, $x$ are the inputs, $\mathcal{L}_{\text{task}}$ is the loss associated with the task (e.g. cross-entropy for text-classification), $\mathcal{L}_{\text{logit}}$ is the KL-divergence between output logits as described in [16], and $\mathcal{L}_{\text{token}}$ is our token-level distillation loss. Hidden tensors passed between consecutive transformer layers are of constant shape $\mathbf{H} \in \mathbb{R}^{B \times seq \times H}$, where $B$ stands for the batch-size, $seq$ for the sequence length, and $H$ for the hidden size defined by the model architecture. This tensor can be interpreted as a collection of $B \times seq$ vectors $\mathbf{h} \in \mathbb{R}^H$, each carrying intermediate model representations of input tokens $x$. We define the loss $\mathcal{L}_{\text{token}}$ as an Euclidean distance $\Delta$ between vectors $\mathbf{h}$ corresponding to each non-padded token in the input sequence, averaged over all unpruned layers. Formally, for a layer $k$, it is defined as

$$\mathcal{L}^k_{\text{token}} = \frac{1}{\sum_{j=1}^{B \times seq} \mathbb{1}[j \notin \mathbf{P}]} \sum_{j=1}^{B \times seq} \mathbb{1}[j \notin \mathbf{P}] \cdot \Delta(\mathbf{h}^{\theta_s}, \mathbf{h}^{\theta_t}), \tag{6}$$

where $\mathbf{P}$ stands for the set of padding tokens. This formulation encourages the student model to generate vector representations for each token that are similar to those produced by the teacher model. In Appendix B, we present ablation studies and comparisons for ZipLM and CoFi, with and without their respective distillation objectives.

## 4 Experiments

**Setup.** Given a pre-trained model, a dataset, and a set of desired speedups in a target inference environment, we iteratively fine-tune and prune the model in a structured way such that in the end we obtain a set of accurate compressed models, one for each speedup target. We consider pruning of the standard BERT$_{\text{base}}$ and BERT$_{\text{large}}$ architectures, evaluating on dev-sets of established benchmarks: SQuADv1.1 [42], and a subset of GLUE [54] tasks: SST-2 [48], QNLI [54], MNLI [57], and QQP [46], selected to match publicly-available checkpoints from prior work. For a precise comparison to prior work [59], our inference environment is a single NVIDIA V100 16GB GPU, batch size of 128, and sequence lengths of 384 and 128 for SQuAD and GLUE tasks, respectively. In addition to encoder-based BERT models, we also consider pruning of the decoder-based GPT2 model on the OpenWebTextCorpus [10], for which we consider two inference environments: pruning for throughput (batch-size=16, sequence-length=1024), and pruning for latency (batch-size=1, a set of prompts with varying lengths). For illustration, our pipeline is depicted in Figure 1. In Appendix H and I, we report exact values for all results, as well as hyper-parameters for reproducibility.

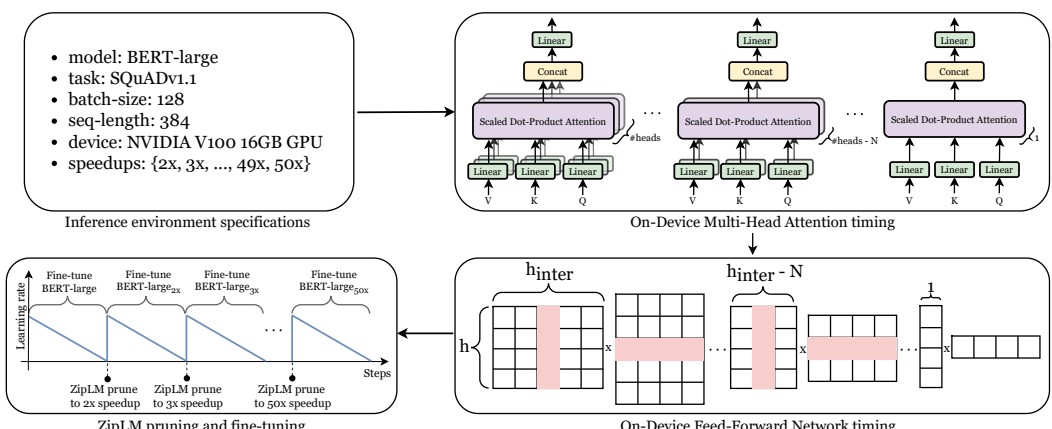

Figure 1: Illustration of the ZipLM pipeline: 1) inference specifications, 2) runtime benchmarking of candidates for pruning, 3) gradual structured pruning until all speedup targets are met.

**Baselines.** In the *gradual pruning* setting, we explore the performance of ZipLM pruning of BERT- and GPT2-family models, across a wide range of inference speedup targets, ranging from 2x to 15x, in unit increments. This allows us to compare the effectiveness of our approach against a diverse set of structured pruning and distillation-based techniques, including state-of-the-art CoFi pruning, competitive Block Movement Pruning, and distillation approaches including TinyBERT, DistilBERT,

DistilGPT2, MobileBERT, MiniLM, and DynaBERT. Additionally, we include comparisons with other relevant methods. For fairness, we follow [59] and report TinyBERT and DynaBERT results without data augmentations. In the *post-training/one-shot* setting, which does not allow retraining, we demonstrate that ZipLM outperforms the prior state-of-the-art approach of [26]. We evaluate inference speedups of all models in the same environment, unless the models are not publicly available, in which case we report speedups from their respective papers. We refer to ZipLM compressed BERT models as ZipBERT, and to ZipLM compressed GPT2 models as ZipGPT2.

## 4.1 Gradual Structured Pruning

**BERT$_{base}$ results.** In Figure 2 we compare structured compression methods on the SQuADv1.1 task. ZipLM outperforms both CoFi and TinyBERT, prior state-of-the-art techniques, by 3 points in the F1 score at the same speedup factor, while at the same F1 score it is able to improve inference speedups by at least 60%. In Figure 3, we extend this comparison to a subset of GLUE tasks and provide an exhaustive overview of various structured compression techniques. Results on the other four remaining GLUE tasks are provided in Appendix Figure 7. As can be observed, distillation-based methods usually provide either one or a few structurally-compressed models, due to the massive costs associated with training from scratch for each new model. Relative to the most competitive approaches, such as TinyBERT, CoFi, and MiniLM, ZipLM provides consistent improvements in terms of both, accuracy and speedup, while providing guarantees for each compressed model in terms of the expected speedup in the target inference environment. Interestingly, on tasks like QQP and SST-2, ZipLM is able to compress the BERT$_{base}$ model up to 6x and 10x speedups, respectively, while maintaining the accuracy of the *dense* model. In Appendix D, we provide additional comparisons against CoFi on test-set results from the official GLUE evaluation server.

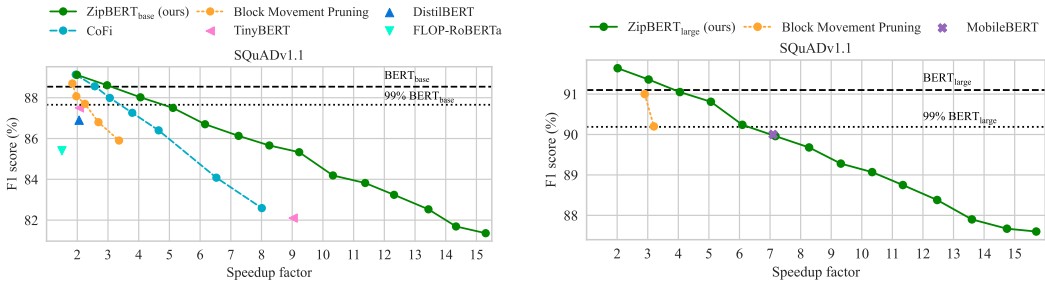

Figure 2: Structured compression of BERT$_{base}$ (left) and BERT$_{large}$ (right) on the SQuADv1.1 task. Dashed horizontal lines represent full and 99% accuracy recovery of the uncompressed model.

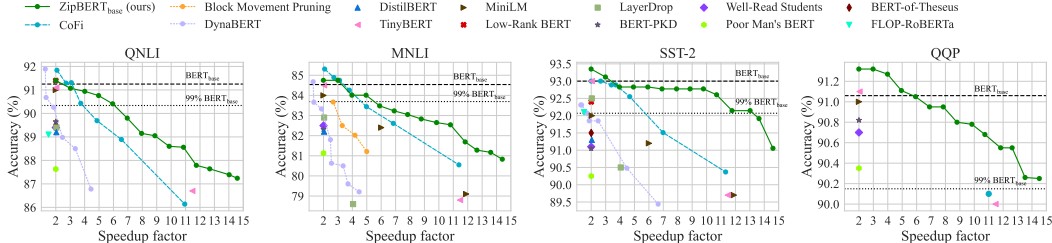

Figure 3: Structured compression of BERT$_{base}$ on QNLI, MNLI, SST-2, and QQP tasks. Dashed horizontal lines represent full and 99% accuracy recovery of the uncompressed model.

**BERT$_{large}$ results.** To verify that our approach does not pertain only to the BERT$_{base}$ model, we apply ZipLM structured pruning to the 3x larger BERT$_{large}$ model on the SQuADv1 task. In this setup, we compare against the only two approaches that attempted to structurally compress this larger model, Block Movement Pruning and distillation-based MobileBERT. As can be seen in Figure 2, ZipLM is able to compress BERT$_{large}$ up to 4x faster inference while maintaining the F1 score of the uncompressed model. At the same F1 score as the fastest Block Movement Pruning model (3x), ZipLM doubles the inference speedup (6x). A result worth emphasizing is that ZipLM is even able to match the performance of the highly optimized MobileBERT model by simply compressing the baseline BERT architecture, without the many additional optimizations and custom components

used by MobileBERT. Specifically, some of the module- and operator-level optimizations used by MobileBERT include: bottleneck structures and carefully-balanced self-attention and feed-forward modules, embedding layer factorization, a bespoke closed-source teacher model, replacement of `LayerNorm` layers with lower-latency `NoNorm` layers, and replacement of `GELU` activation functions with `ReLU` activations.

**99% recovery.** The MLPerf Benchmark [36] targets recovery of >99% of the baseline accuracy. At this industry-defined threshold, ZipLM models set new state-of-the-art performance across all of the considered datasets with the following BERT$_{base}$ inference speedups: 5x on the SQuADv1 task, 6x on QNLI and MNLI, and, surprisingly, 13x and 15x on SST-2 and QQP, respectively. When compressing BERT$_{large}$ on the SQuADv1 task, ZipLM produces a 6x faster model at 99% recovery.

**GPT2 results.** To validate that our approach does not only apply to encoder-based models, we apply ZipLM structured pruning to the decoder-based GPT2 model. In addition to this, to further demonstrate the inference-awareness property of our approach and its importance for real-world applications, we consider two different regimes: pruning for throughput and pruning for latency. An example application for the former regime is a server-side deployment where the model processes many queries at the same time, while an application for the latter regime is a text-generation scenario where the model is used in an online fashion to auto-complete user's text.

For a fair comparison, we follow the DistilGPT2 setup [44] and prune the 124M parameters GPT2 variant on the OpenWebTextCorpus dataset, followed by *zero-shot* evaluations, without any fine-tuning, on the test-split of the WikiText [37] dataset. Because of the enormous vocabulary size, the maximum achievable speedup in the throughput regime for this model is roughly 3.5x. Thus, we run ZipLM pruning to 1.5x, 2x, 2.5x, and 3x speedup targets. For the latency regime, we report the median time to process sequences of various lengths when generating text with Top-K sampling [4]. In Table 1, we present zero-shot evaluations of the uncompressed GPT2 model which serves as a baseline relative to the competing DistilGPT2 approach, and four variants of our ZipLM pruned GPT2. In the pruning for throughput scenario, at similar speedup and decoder size (1.6x-vs-1.5x and 42.5M-vs-47.3M), ZipGPT2 achieves significantly lower perplexities relative to DistilGPT2. Further, at slightly better (lower) perplexities, ZipGPT2 reduces the decoder size from 42.5M to only 26.5M parameters (60% reduction) and improves speedup from 1.6x to 2.1x (30% faster). In the pruning for latency scenario, at a similar speedup of 1.9x-vs-2.0x, ZipGPT2 reduces the decoder size by 3M params while providing almost 2 points improvement in the zero-shot perplexity.

Table 1: Zero-Shot perplexity (PPL) of compressed GPT2 in two regimes: pruning for throughput and pruning for latency. *GPT2 was trained by OpenAI [41] on a much larger closed-source dataset and for significantly longer. The only direct comparison is between DistilGPT2 and ZipGPT2.

Table 2: One-shot (post-training) structured pruning of BERT$_{base}$ on three downstream datasets and two speedup targets.

| Model | Pruning for throughput | | | Pruning for latency | | |
|---|---|---|---|---|---|---|
| | Speedup | Decoder size | Wiki Text-103 PPL ↓ | Speedup | Decoder size | Wiki Text-103 PPL ↓ |
| GPT2* | 1.0x | 85.0M | 28.5 | 1.0x | 85.0M | 28.5 |
| DistilGPT2 | **1.6x** | **42.5M** | **43.0** | **1.9x** | **42.5M** | **43.0** |
| ZipGPT2 (ours) | **1.5x** | **47.3M** | **35.4** | **1.6x** | **48.7M** | **37.8** |
| | **2.1x** | **26.5M** | **41.5** | **2.0x** | **39.2M** | **41.2** |
| | 2.7x | 14.0M | 50.4 | 2.2x | 26.6M | 49.0 |
| | 3.3x | 5.7M | 72.1 | 2.5x | 20.7M | 55.0 |

| Speedup | Kwon et al. [26] | ZipBERT base |
|---|---|---|
| SQuAD, F1 | | |
| 1.5x | 86.2 | **87.1** |
| 2.0x | 76.5 | **84.1** |
| QQP, acc. | | |
| 1.5x | 89.5 | **89.7** |
| 2.0x | 83.9 | **84.8** |
| MNLI, acc. | | |
| 1.5x | 82.8 | **83.0** |
| 2.0x | 78.1 | **78.2** |

## 4.2 On the Importance of Inference-Awareness

**Depth vs. width pruning.** A particularly interesting illustration of the importance of inference-awareness in the pruning algorithm is given by our GPT2 models running directly in the PyTorch-HuggingFace framework, which can be used in two different modes: batch-prediction (throughput-constrained) and text-generation (latency-constrained). For the former, inputs are typically large, and shrinking weight matrices is an effective way to achieve speedups. However, for the latter, the inputs are much smaller, and the size of weight matrices is no longer the primary bottleneck.

In this scenario, the only way to achieve substantial speedups is to completely drop some modules, which prior methods cannot account for as they solely optimize for overall model sparsity. However, with ZipLM, runtime measurements from the target inference environment guide pruning decisions, allowing it to learn the best way to compress the model for an optimal speedup-accuracy trade-off. Our GPT2 compression results in Table 1 clearly illustrate and support these statements. Even though pruned for the same speedup target, the final architectures of ZipGPT2 models are drastically different. For the throughput-constrained scenario, the model's depth was preserved but the matrix dimensions were significantly reduced (roughly by a factor of 10) making the corresponding multiplications with large input tensors much faster. In contrast, for the latency-constrained scenario, the model's width (shapes of weight matrices) was mostly preserved but the depth was shrunk almost by a factor of 4, making the forward pass with small inputs faster by reducing the effective number of modules.

**Inference device capabilities.** Incorporating capabilities of the inference device is another important aspect for effective structured pruning which prior methods do not account for as they solely optimize for higher sparsities. As noted in Section 3.2, this reflects in larges discrepancies between speedups obtained on different devices, e.g. a compressed model with 12x speedup on a V100 is only 5x faster on an A100 GPU. This arises because the A100 GPU is significantly more powerful and thus faster on the dense model; at the same time, it is highly underutilized for small matrices, which significantly limits the speedups for very high sparsity. To illustrate this, we have measured the speedup from reducing the MLP size for both GPU types (see Table 3). As can be seen, pruning to ≈90% sparsity (3072 → 302) gives ≈7x speedup on a V100 but only ≈3x speedup on an A100. Such differences are automatically captured by ZipLM, where pruning for sparsity is replaced by pruning for speedup.

**Pruning for speedup vs. pruning for sparsity.** In Figure 4 we compare results with ZipLM pruning when the target for pruning is sparsity (like prior approaches) and when the target for pruning is speedup (the ZipLM approach). Pruning for speedup brings significant improvements, up to 10 points, especially at higher speedups where inference-awareness is very important as the algorithm does not remove components that do not bring any further speed and therefore helps preserving accuracy.

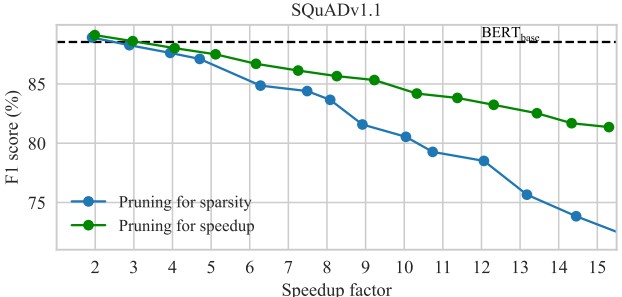

Figure 4: Ablation study for the impact of the pruning target: pruning for sparsity (like prior approaches) versus pruning for speedup (the ZipLM approach).

Table 3: Speedups from shrinking the intermediate size of MLPs in the FFN section of a Transformer layer, on different GPUs.

|  | Speedup | |
| --- | --- | --- |
| MLP size | V100 | A100 |
| 3072 | 1.0x | 1.0x |
| 1814 | 1.6x | 1.1x |
| 1322 | 2.0x | 1.4x |
| 302 | 6.9x | 3.1x |
| 130 | 11.8x | 4.4x |
| 76 | 13.1x | 4.4x |
| 33 | 14.8x | 4.4x |

### 4.3 Post-training/One-shot Structured Pruning

We now study the performance of ZipLM when applied purely in *one-shot*, without any retraining. In this setting, we compare against the state-of-the-art method of Kwon et al. [26] which combines several heuristics: Fisher-based mask search, mask rearrangement, and mask tuning. Instead of heuristics, our pruning framework utilizes direct end-to-end loss information to find the optimal sparsity configuration. During the warm-start phase, [26] utilizes a diagonal Fisher matrix to estimate the significance of heads and filters, which discards correlations caused by off-diagonal elements. Although the approach attempts to address this limitation by approximating correlations within a single layer, it will not capture global dependencies. Furthermore, the weights are adapted for layer-wise reconstruction at the very end of the compression step, whereas our method does it continuously during the pruning (please see Section 4 for the significance of doing this). For a fair comparison, we apply the authors' own implementation in latency-constrained mode on the exact same model weights. Table 2 presents results on several datasets and speedups, showing that ZipLM is even more accurate than the approach designed and optimized specifically for the post-training/one-shot pruning.

**Sensitivity to calibration data.** Additionally, we have found that ZipLM is very robust to the amount of calibration data. In Table 4 we present a sensitivity analysis with respect to the number of calibration samples. We one-shot prune BERT$_{\text{base}}$ on the SQuADv1.1 task for two speedup targets: 1.5x and 2.0x. In this setup, we compare results against Kwon et al. [26], which uses 2048 samples by default. As can be seen from the table, ZipLM outperforms prior state-of-the-art starting at only 32 samples. As we increase the number of samples, the results improve, up to 2 points in F1 score.

## 5 Discussion and Extensions

**CPU as an LLM-inference environment.** In Section 4 we have focused on various GPU-based inference environments as it enabled us to conduct fair comparisons against prior structural compression techniques. However, CPUs present another compelling inference environment focused on edge deployment of LLMs. Therefore, we target the recently proposed compound compression pipeline of [24], which involves three steps: structured pruning, unstructured pruning, and quantization. We replace their structured pruning approach based on layer dropping with ZipLM. As a result, at full accuracy recovery, we are able to improve speedup from 3x to 13x, and at the largest compression ratio from 30x to 50x. Due to space constraints, we provide full results in Appendix A.

**Computational efficiency.** Relative to distillation-based methods, structured pruning is an order of magnitude more efficient in terms of GPU hours due to the massive costs associated with pretraining from scratch for each compressed model [59, 51, 21]. For efficiency comparisons to CoFi, we consider the task of producing a full family of compressed BERT$_{\text{base}}$ models with speedup targets ranging from 2x to 15x. In this setup, ZipLM requires only 115 epochs in total, whereas CoFi would require 560 epochs. Therefore, ZipLM is *4.87 times more efficient* than CoFi. In terms of end-to-end runtime, ZipLM produces the entire family of compressed BERT$_{\text{base}}$ models on a single RTX A6000 GPU in ∼35 hours on larger datasets (e.g. MNLI) and only ∼10 hours on smaller ones (e.g. SST2). Finally, it is worth emphasizing that we have not taken into account the cost of hyper-parameter tuning in the above comparisons, but that this is very favorable to ZipLM: it uses a single set of hyper-parameters to produce an entire family of compressed models while other methods require hyper-parameter tuning for each model independently.

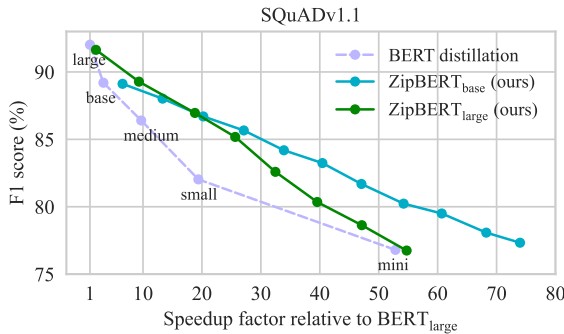

Figure 5: Scaling laws of structured pruning vs. distillation on the standard BERT architecture.

Table 4: Sensitivity to the number of calibration samples.

| Method | Num samples | F1 score at | |
| --- | --- | --- | --- |
| | | 1.5x | 2.0x |
| ZipLM | 4 | 82.3 | 48.4 |
| | 32 | **86.8** | **82.6** |
| | 128 | **86.8** | **83.6** |
| | 512 | **86.8** | **84.1** |
| | 2048 | **87.1** | **84.1** |
| | 4096 | **87.6** | **84.7** |
| Kwon et al. | 2048 | 86.2 | 76.5 |

**Scaling laws for structured pruning.** To further understand the accuracy-speedup trade-offs, we run ZipLM on larger speedup ratios, up to **55x for BERT$_{\text{large}}$** and **75x for BERT$_{\text{base}}$**. To the best of our knowledge, this is the first result in literature demonstrating that such extreme compression ratios are achievable with structured pruning without model collapse. In Figure 5, we compare these results against distillation-based downscaling of the BERT architecture [52]. The results clearly demonstrate that each of the pruned models, based either on BERT$_{\text{large}}$ or BERT$_{\text{base}}$, significantly outperforms comparable pre-trained variants. An emergent behavior that can be observed is that structurally pruned models tend to follow a linear scaling law, meaning that the accuracy decreases linearly with the increase of the speedup ratio, at a slope given by the original model. Fitting linearly via least squares produces the following expressions for the accuracy-speedup relationship: $\texttt{F1}_{\texttt{large}} \approx 92.1 - 0.3 \times \texttt{speedup}_{\texttt{large}}$, and $\texttt{F1}_{\texttt{base}} \approx 90.3 - 0.6 \times \texttt{speedup}_{\texttt{base}}$. Thus, the rate of decrease in accuracy for BERT$_{\text{base}}$ is twice as large as that of BERT$_{\text{large}}$, which can be attributed to the presence of more redundant representations in the larger model, making it more resilient to pruning. In Appendix G we provide additional analysis of the structure of pruned models.

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

## A Compound Compression for Edge Deployment

Deploying large language models in edge environments requires running inference on low-power devices such as CPUs. Therefore, we follow the compound compression approach from [24] which bundles together structured, unstructured pruning, and quantization for efficient inference on CPUs. We start with ZipLM structurally pruned models, and apply on top the state-of-the-art oBERT unstructured pruning method [24] to 80% sparsity. After structured and unstructured pruning, we apply quantization-aware-training (QAT) [20] to quantize FP32 weights into INT8 representations. We benchmark these compound compressed models by running inference in the DeepSparse [39] engine, on a *single-core* of Intel Cascade Lake CPU. In this setting, we compare our results against the compound compression pipeline of [24] which applies layer dropping as a form of structured pruning. As can be seen from Figure 6, when we substitute layer dropping with a principled structured pruning via ZipLM, the resulting compound compressed models achieve very competitive latency-vs-accuracy performance in the edge-inference regime. At full accuracy recovery, ZipLM improves the speedup from 3x to 13x, while at the largest compression ratio ZipLM improves the speedup from 30x to 50x.

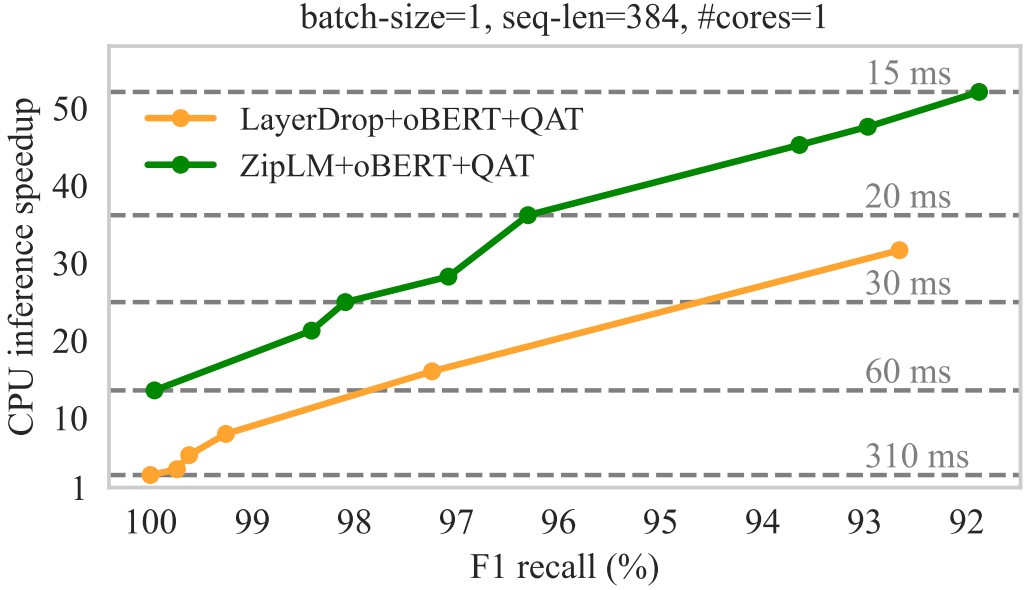

Figure 6: Improvements in CPU-inference speedups for compound compressed BERT$_{base}$ models on the SQuADv1.1 task when ZipLM is used for structured pruning. End-to-end latency indicated by the dashed line.

## B Ablation Studies

In Table 5, we present ablation results for ZipLM and CoFi, with and without their respective layer-wise distillation techniques. ZipLM outperforms CoFi in all tasks when both methods use distillation, and in three out of four when distillation is not used. For example, ZipLM outperforms CoFi with a significant 3 point increase in F1 score on the SQuAD task in both setups. Furthermore, when comparing ZipLM results with and without layer-wise distillation, it can be observed that benefits are pronounced for low data tasks, where accuracy improvements reach up to 2 points.

## C Additional GLUE results

Due to space constraints, in Figure 3 we present results only on four GLUE tasks. Therefore, for completeness, in Figure 7 we present results on the remaining four GLUE tasks, namely: CoLA, MRPC, STS-B, and RTE. Results show the same trends as the other four GLUE tasks, with large improvements for ZipLM, especially at higher compression rates.

Table 5: Comparison of ZipLM and CoFi dev-set results, with and without layer-wise distillation.

| | SST-2 acc. | QNLI acc. | MNLI m-acc. | SQuAD F1 |
|---|---|---|---|---|
| CoFi | 90.4 | 86.1 | 80.6 | 82.6 |
| ZipBERT$_{\text{base}}$ | **91.7** | **88.6** | **81.7** | **85.7** |
| CoFi w/o $\mathcal{L}_{\text{layer}}$ | **91.1** | 85.1 | 79.7 | 82.5 |
| ZipBERT$_{\text{base}}$ w/o $\mathcal{L}_{\text{token}}$ | 89.2 | **86.5** | **81.2** | **85.7** |

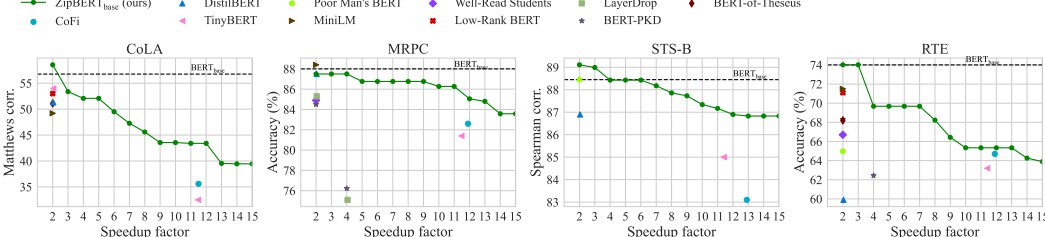

Figure 7: Structured compression of BERT$_{\text{base}}$ on CoLA, MRPC, STS-B, and RTE tasks. Dashed horizontal lines represent full accuracy recovery of the uncompressed model.

## D   Additional Validation

Evaluating and comparing compressed models on the development set (dev-set) is standard practice, as it enables comparisons with off-the-shelf results from the literature. However, an implicit assumption behind such comparisons is that all methods tune their hyper-parameters only on a subset of the dev-set before evaluating and reporting results on all samples, which is not always the case. Moreover, specifically-tuned hyper-parameters can lead to large performance differences, especially when compressing LLMs [23]. To ensure that there is no such "overfitting" on the dev-set, in Table 6 we compare ZipLM against the prior state-of-the-art CoFi approach on *unseen test-set*, obtained by submitting predictions to the official GLUE evaluation server. The results show consistent improvements over CoFi, on both dev- and test-sets.

## E   Latency table used for ZipLM pruning

As described in Section 3.2, we record the time to run an attention block, including all overheads, with $0, \ldots, N_{\text{heads}} - 1$ heads pruned (pruning everything with runtime 0 is also considered) and similarly for the fully-connected block with the intermediate dimension shrunk by a factor of $0.9^i$, for $i = 0, \ldots, 42$; in relative steps of $10\%$ up until $\approx 99\%$ sparsity, following [5]. In Table 7 we present an example of such a latency table used in ZipLM pruning approach.

## F   Speedup Evaluations

As shown in Figure 1, ZipLM is based on measuring runtimes of higher-level modules, such as attention heads and fully connected matrices, rather than low-level operators. This makes our

Table 6: Dev- and test-set comparison of ZipBERT$_{\text{base}}$ and CoFi models with comparable speedups.

| | dev-set | | test-set | |
|---|---|---|---|---|
| | CoFi | ZipBERT$_{\text{base}}$ | CoFi | ZipBERT$_{\text{base}}$ |
| QNLI, acc. | 86.1 | **88.6** | 85.8 | **88.4** |
| SST-2, acc. | 90.4 | **91.7** | 88.2 | **91.8** |
| MNLI, m-acc. | 80.6 | **81.7** | 80.7 | **81.9** |
| MNLI, mm-acc. | 80.7 | **82.0** | 79.9 | **80.6** |
| SQuAD, F1 | 82.6 | **85.7** | N/A | N/A |

Table 7: An example of the latency table used by the ZipLM pruning approach.

| Intermediate size | Latency (ms) | Number of heads | Latency (ms) |
|---|---|---|---|
| 3072 | 11.9 | 12 | 7.9 |
| 1814 | 7.4 | 10 | 6.7 |
| 1322 | 5.8 | 8 | 5.8 |
| 302 | 1.6 | 6 | 4.4 |
| 130 | 1.0 | 4 | 3.2 |
| 76 | 0.9 | 2 | 1.9 |
| 33 | 0.7 | 0 | 0 |

Table 8: Comparison of target (desired) inference speedups with achieved (on-device measured) speedups obtained with our ZipLM pruning approach.

| BERT$_{base}$ on SQuADv1.1 | | | BERT$_{large}$ on SQuADv1.1 | | |
|---|---|---|---|---|---|
| Target speedup | Achieved speedup | Deviation | Target speedup | Achieved speedup | Deviation |
| 2 | 1.98 | -1.00% | 2 | 2.01 | +0.50% |
| 4 | 4.05 | +1.25% | 4 | 4.05 | +1.25% |
| 6 | 6.16 | +2.67% | 6 | 6.09 | +1.50% |
| 8 | 8.25 | +3.12% | 8 | 8.27 | +3.37% |
| 10 | 10.36 | +3.60% | 10 | 10.33 | +3.30% |
| 12 | 12.31 | +2.58% | 12 | 12.46 | +3.83% |
| 14 | 14.33 | +2.35% | 14 | 14.74 | +5.28% |

approach independent of underlying optimizations in different inference engines and frameworks, which usually perform further optimizations such as operator-fusion. Our runtime lookup table contains information about the runtime of a Transformer layer with different numbers of attention heads, and various dimensions of the fully connected matrices. This implies that we measure runtimes of a layer with 12 heads, 11 heads, 10 heads, and so on, as well as the runtimes of fully connected matrices with hidden sizes ranging from 3072 to 0. We utilize this information to guide pruning decisions.

To fully validate the ability of ZipLM to compress the model while satisfying desired speedup constraints via the described approach, we provide the timing results in Table 8, comparing the desired (target) speedup and the achieved (measured) speedup for different models.

As can be seen from the Table 8, the deviation between the desired (target) and the achieved (measured) speedup is at most 5.28%. This confirms that our approach indeed provides reliable runtime information to guide the pruning decisions.

# G    Structure of Pruned Models

Through a comprehensive examination of ZipLM pruned BERT models across all datasets considered in Section 4, we aim to identify trends in the pruning of key components of the Transformer layer, namely attention heads and intermediate size, needed to achieve a specific speedup target. As illustrated in Figure 8, we observe that the intermediate size is pruned at a higher rate relative to attention heads, which aligns with the fact that the intermediate size dictates the dimensions of the two large linear layers in the feed-forward part of the Transformer block. For instance, to attain a 2x speedup, roughly 60% of the intermediate size and 40% of the attention heads need to be removed. Additionally, in Figure 9, we visualize the entire encoder size needed to reach a specific speedup target. Interestingly, we find that 15x faster models retain on average only 2% of intermediate size and 6% of attention heads which amounts to only 2.9M parameters overall, while at the same time recovering more than 95% of the uncompressed model's accuracy (see Figure 3).

Additionally, in Figures 10, 11, 12, 13 we visualize the number of remaining heads and intermediate size across all Transformer layers and various speedup targets on a subset of GLUE datasets.

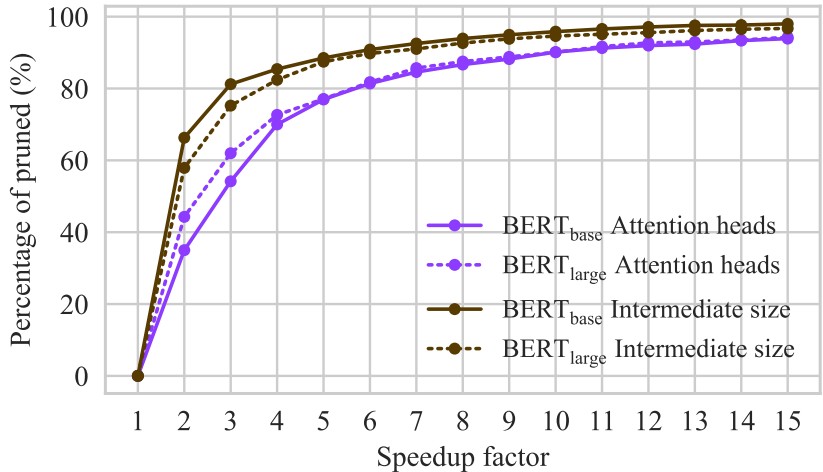

Figure 8: Percentage of pruned attention heads and intermediate size to reach a specific speedup target with ZipLM.

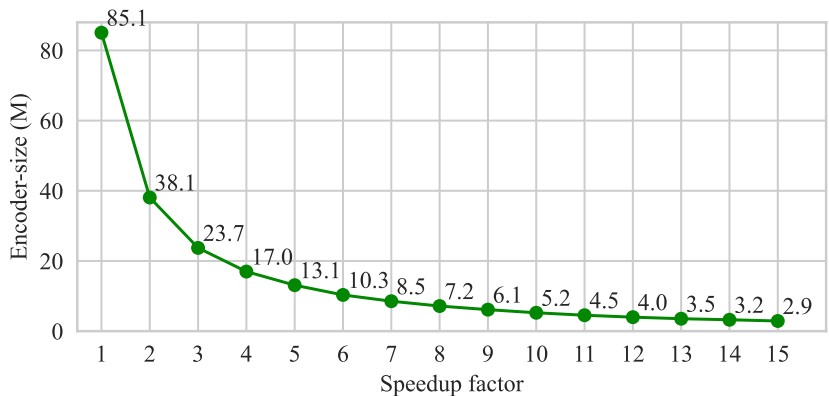

Figure 9: Encoder size vs. speedup factor of ZipLM pruned $BERT_{base}$ models, averaged over all considered datasets in Section 4.

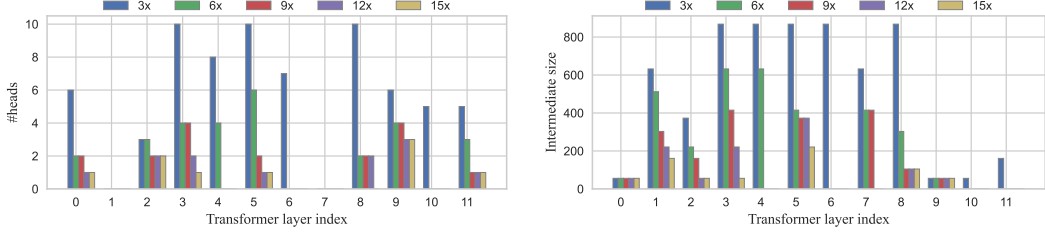

Figure 10: Remaining number of attention heads and intermediate size across all layers of the ZipLM compressed $BERT_{base}$ model at various speedups and MNLI dataset.

# H   Experiments - Additional Results

In Table 9 we report accuracy and model size of ZipLM pruned models visualized in Section 4, in Figures 2 and 3.

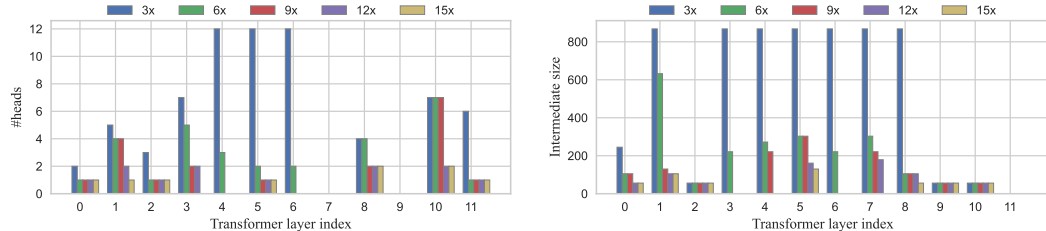

Figure 11: Remaining number of attention heads and intermediate size across all layers of the ZipLM compressed BERT_base model at various speedups and QNLI dataset.

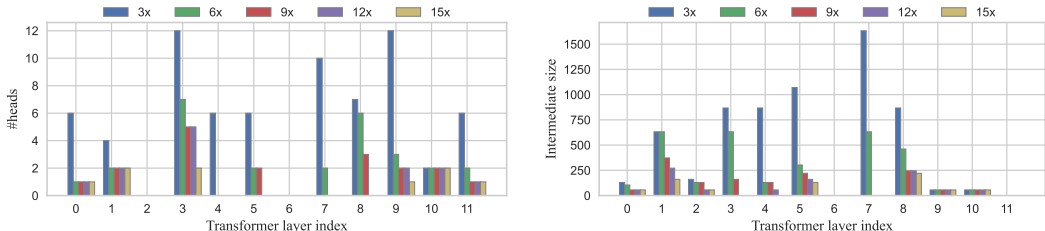

Figure 12: Remaining number of attention heads and intermediate size across all layers of the ZipLM compressed BERT_base model at various speedups and QQP dataset.

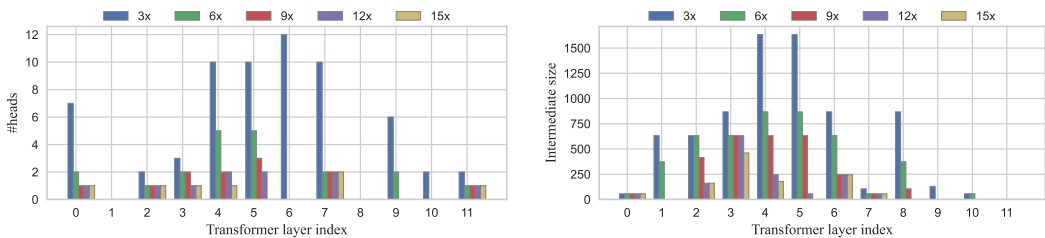

Figure 13: Remaining number of attention heads and intermediate size across all layers of the ZipLM compressed BERT_base model at various speedups and SST-2 dataset.

# I   Hyper-parameters for Reproducibility

To facilitate reproducibility, we conduct experiments in the open-source Transformers library [58], and use publicly available datasets [28]. We plan to open-source our entire framework which supports one-shot and gradual structured pruning via SparseML [25], making it very easy to experiment with other models and datasets. In addition to our code, we plan to open-source all of our compressed models via the popular HuggingFace Hub. In Table 10 we report hyper-parameters used to produce our ZipLM pruned models in Section 4. Because of the excessive memory overhead, we don't make use of any kind of knowledge distillation when pruning the GPT2 model. Following insights from DistilGTP2, we hypothesize that this can further improve our results. We follow [22] and disable dropout regularization while pre-training ZipGPT2 models at OpenWebTextCorpus dataset.

# J   Broader Impact and Limitations

Our results contribute to the line of work on efficient language models. Thus, it should help reduce the energy and monetary cost of inference over such models, and allow them to be used without access to powerful hardware. While this is a mainly positive outcome, it also reduces the cost of employing these models for detrimental purposes, such as spam generation. Thus, this significant cost reduction for inference should also be seen as further motivation for methods to ensure safe usage of these models, such as watermarking or alignment.

Table 9: Accuracy and model size for ZipLM pruned models in Section 4.

| | BERT$_{base}$ | | | | | | | | | | BERT$_{large}$ | |
| | QNLI | | MNLI | | SST2 | | QQP | | SQuADv1 | | SQuADv1 | |
| Speedup | Acc. | Encoder size (M) | Acc. | Encoder size (M) | Acc. | Encoder size (M) | Acc. | Encoder size (M) | F1 | Encoder size (M) | F1 | Encoder size (M) |
|---|---|---|---|---|---|---|---|---|---|---|---|---|
| 2x | 91.4 | 38.0 | 84.8 | 38.5 | 93.4 | 38.7 | 91.3 | 37.8 | 89.1 | 37.3 | 91.6 | 141.1 |
| 3x | 91.1 | 23.8 | 84.8 | 23.5 | 93.4 | 24.1 | 91.3 | 23.8 | 88.6 | 23.4 | 91.4 | 88.3 |
| 4x | 90.9 | 16.9 | 84.0 | 17.1 | 93.0 | 17.2 | 91.3 | 16.8 | 88.0 | 16.8 | 91.1 | 63.1 |
| 5x | 90.8 | 12.5 | 84.0 | 13.5 | 93.0 | 13.5 | 91.1 | 13.0 | 87.5 | 13.0 | 90.8 | 48.5 |
| 6x | 90.4 | 9.5 | 83.5 | 10.5 | 93.0 | 11.0 | 91.1 | 10.2 | 86.7 | 10.4 | 90.2 | 39.1 |
| 7x | 89.8 | 8.0 | 83.2 | 8.8 | 93.0 | 9.0 | 90.9 | 8.1 | 86.1 | 8.7 | 89.9 | 32.7 |
| 8x | 89.2 | 6.4 | 83.1 | 7.5 | 93.0 | 7.6 | 90.9 | 6.8 | 85.7 | 7.5 | 89.7 | 27.5 |
| 9x | 89.1 | 5.7 | 82.8 | 6.3 | 93.0 | 6.7 | 90.8 | 5.8 | 85.3 | 6.2 | 89.3 | 23.8 |
| 10x | 88.6 | 4.9 | 82.7 | 5.4 | 93.0 | 5.7 | 90.8 | 4.9 | 84.2 | 5.3 | 89.1 | 20.9 |
| 11x | 88.6 | 4.0 | 82.5 | 4.7 | 92.7 | 4.9 | 90.7 | 4.3 | 83.8 | 4.7 | 88.8 | 18.4 |
| 12x | 87.8 | 3.6 | 81.7 | 4.1 | 91.7 | 4.2 | 90.6 | 4.1 | 83.2 | 4.0 | 88.4 | 16.4 |
| 13x | 87.6 | 3.2 | 81.3 | 3.5 | 91.7 | 3.8 | 90.6 | 3.7 | 82.5 | 3.4 | 87.9 | 14.9 |
| 14x | 87.4 | 2.8 | 81.2 | 3.3 | 91.7 | 3.6 | 90.3 | 3.3 | 81.7 | 3.2 | 87.7 | 13.7 |
| 15x | 87.2 | 2.6 | 80.8 | 2.9 | 90.7 | 3.2 | 90.3 | 2.9 | 81.4 | 2.9 | 87.6 | 12.5 |

Table 10: Hyper-parameters used for gradual ZipLM runs in Section 4.

| | BERT$_{base}$ | BERT$_{large}$ | GPT2 |
|---|---|---|---|
| batch-size | 16 SQuADv1 | 32 GLUE | 128 |
| max-seq-length | 384 SQuADv1 | 128 GLUE | 1024 |
| finetune before pruning | 3 epochs | | 50k steps |
| finetune in-between pruning steps | 8 epochs | 10 epochs | 2 epochs |
| LR schedule in-between pruning steps | linear decay | | linear decay |
| initial LR | 8e-5 | 5e-5 | 1e-3 |
| #calibration samples | 2048 | | 512 |
| speedup-targets | {2, 3, 4, 5, ..., 15}x | | {1.5, 2, 2.5, 3}x |
| knowledge distillation $\lambda_1$ | 0 | | 1.0 |
| knowledge distillation $\lambda_2$ | 1.0 SQuADv1 | 0.5 GLUE | 0 |
| knowledge distillation $\lambda_3$ | 0.0 SQuADv1 | 0.5 GLUE | 0 |
| weight-decay | 0.03 | 0.05 | 0 |

As any academic study, our work is not without its limitations. All of our benchmarks are focused on English-language datasets and therefore our results do not provide insights into compression effects for low-data languages. Unfortunately, this limitation is inherent to all of the existing works on compression due to the lack of standardized benchmarks. Given that our structured pruning approach relies on a small sample of calibration data to perform pruning decisions, we hypothesize that our approach should be able to provide satisfying results in the low-data setup as well. At the moment we do not have data to support these claims, but we see it as an opportunity for future work.

