# OpenReview forum: "ZipLM: Inference-Aware Structured Pruning of Language Models"
_NeurIPS.cc/2023/Conference — NeurIPS 2023 poster_

### Official Review · Reviewer_5zkW · 2023-07-05

**Soundness:** 3 good
**Presentation:** 3 good
**Contribution:** 3 good
**Rating:** 6
**Confidence:** 3

**Summary:**

This paper proposed a novel structured compression approach for the inference of LLM, which reduces the model size by removing entire sub-components, like rows or columns from the model’s weight matrices. Based on their algorithm, they manage to drop out some attention heads and shrink the size of FFN layers. Besides, the proposed method considers the runtime speedup while deciding the sparsity of the pruning, which leads to a better trade-off between the loss and the performance.

**Strengths:**

- This work comes up with a new structured pruning method for the transformer models and achieved reasonable speedup in the experiments with multiple models.
- The theoretical analysis is solid.

**Weaknesses:**

- The performance of the proposed method seems strongly dependent on the data distribution of inference, which may make it not that practical in real-world inference settings where the incoming inputs can vary a lot.
- In the latency-constrained scenario, the authors said "However, for the latter, the inputs are much smaller, and the size of weight matrices is no longer the primary bottleneck.", however, in my opinion, when the batch size is very small (like 1), loading weights from GPU HBM is exactly the bottleneck. Therefore, I guess sparsifying model parameters should lead to reasonable speedups?
- In the throughput-constrained (aka. large batch size), it will be interesting to see how ZipLM could support an even larger batch size since we can take advantage of the memory saved by pruning to hold more inputs. In this way, we may further increase the throughput?

**Questions:**

Please refer to the Weekness

**Limitations:**

The potential social impact is not discussed in the paper.

---

> ### Author Rebuttal · Authors · 2023-08-09
>
> **Question 1: The performance of the proposed method seems strongly dependent on the data distribution of inference, which may make it not that practical in real-world inference settings where the incoming inputs can vary a lot.**
>
> We would like to note that encoder-only models, such as BERT, are designed for per-task finetuning. Similarly, ZipLM enables per-task pruning, to very large speedup ratios. This per-task setting is completely standard in pruning & distillation literature around encoder-only models. Additionally, we also apply ZipLM to decoder-only models, which are typically used for diverse tasks without additional finetuning. Consequently, we perform pruning on a very general dataset (OpenWebText) and evaluate the model in zero-shot fashion on the unseen WikiText dataset.
>
> In terms of sensitivity to different amounts of calibration data, we have performed additional experiments and found that ZipLM is very robust: even with only 32 samples, we outperform the prior state-of-the-art Kwon et al. using 2048 samples, in the one-shot setting. We present results in the table below:
>
> |       Speedup = 1.5x    |   |  $\|$  |  Speedup = 2.0x | |
> | :----------------: | :-------- | :--------------: | :--------------: | :-------------: |
> | Num samples      | F1 score |     $\|$           | Num samples | F1 score |
> | ZipLM, 4         | 82.33    |        $\|$        | ZipLM, 4 | 48.41 |
> | ZipLM, 32        | 86.76    |        $\|$        | ZipLM, 32 | 82.63 |
> | ZipLM, 128       | 86.79    |      $\|$          | ZipLM, 128 | 83.56 |
> | ZipLM, 512       | 86.79    |       $\|$         | ZipLM, 512 | 84.06 |
> | ZipLM, 2048      | 87.08    |       $\|$         | ZipLM, 2048 | 84.14 |
> | ZipLM, 4096      | 87.62    |      $\|$          | ZipLM, 4096 | 84.68 |
> | Kwon et al, 2048 | 86.20     |     $\|$           | Kwon et al, 2048 | 76.50 |
>
> Finally, to ensure that there is no overfitting of hyper-parameters to the validation set, we have also performed test set evaluations on the official GLUE servers in Appendix C: Additional Validation.
>
> **Question 2: In the latency-constrained scenario, the authors said "However, for the latter, the inputs are much smaller, and the size of weight matrices is no longer the primary bottleneck.", however, in my opinion, when the batch size is very small (like 1), loading weights from GPU HBM is exactly the bottleneck. Therefore, I guess sparsifying model parameters should lead to reasonable speedups?**
>
> Yes, structured pruning does, in general, bring speedups for memory-bound applications like small batch-size generative inference as well. What we are referring to in the paper is that for our particular model and inference environment, generative runtime is significantly impacted by overheads (e.g. kernel launches, layer-norms, under-utilization, etc.) as low batch-size matmuls are so fast (on larger models or on low-power devices, this should be much less significant). This means size reduction does not necessarily lead to proportional speedups. ZipLM, however, understands this automatically and accounts for it during pruning. This is what we wanted to illustrate in this part. Nevertheless, we agree that this can be made more clear and will make improvements in the next revision.
>
> **Question 3: In the throughput-constrained (aka. large batch size), it will be interesting to see how ZipLM could support an even larger batch size since we can take advantage of the memory saved by pruning to hold more inputs. In this way, we may further increase the throughput?**
>
> Yes, this is an excellent suggestion, and we therefore ran some benchmarking tests on an 11GB RTX 2080Ti. For each model, dense (1.0x) and ZipLM compressed (4.9x, 9.8x, 14.5x) models, we increase batch-size to the maximum value and evaluate the throughput (number of samples processed per second). The table below presents throughput improvements from increased batch-size for the corresponding speedup targets.
>
> | Speedup | Throughput gain |
> | :-------: | :---------------: |
> | 1.0x    | 1.0x            |
> | 4.9x    | 6.5x            |
> | 9.8x    | 12.0x           |
> | 14.5x   | 17.6x           |
>
> **Question 4: The potential social impact is not discussed in the paper.**
>
> Please note that, following the NeurIPS23 submission guidelines, we have provided a discussion of limitations and impact in the 'Appendix H: Broader Impact and Limitations'.

---

### Official Review · Reviewer_rTQd · 2023-07-05

**Soundness:** 2 fair
**Presentation:** 3 good
**Contribution:** 3 good
**Rating:** 5
**Confidence:** 4

**Summary:**

This paper looks into structure pruning (e.g. neurons, or attention heads) of Transformer based models.
Authors propose to formulate the pruning objective by requiring the output of each pruned layer to be as close as possible to that of an unpruned layer. Then they adopt optimal brain surgeon algorithm to come up with the weights to keep (mask) and the update to such weights. The proposed algorithm prunes each group (e.g. a neuron in a layer) one by one, ensuring that correlated groups are properly accounted for (which is hard to do when choosing a number of neurons to prune in one go).
Additional extensions to the method include inclusion of inference aware criteria into the algorithm for choosing the groups to keep (using pre-computed knowledge of latency table)
Suggested approach can be used for one shot pruning and gradual pruning. Experimental results indicate good performance on attention head pruning, fully connected layers pruning and removing full attention modules.

**Strengths:**

- Really good experimental results
- Well written and easy enough to follow (modulo my comments below)

**Weaknesses:**

- Not very clear novelty
- seems computationally expensive
-  Ablations are missing (is the improvement coming from the fact that you use formulation (1) for each layer? is it from line 155 of the algorithm? Is it the inference awareness inclusion?

**Questions:**

1) How does pruning two consecutive layers happens? First pruning first layer (Algorothm 1) and then pruning the next layer?
If yes you are essentially assuming that layers are independent (so assuming block diagonal structure of the hessian).
2) In your formulation (1) you seem to suggest that X is the input to the layer (whose weights W you are considering to prune). In the original Optimal Brain Surgeon, X is actually based on the gradients (and the global objective is considered, instead of local l2 objective of matching each layer's output). I think this all needs to be made clearer
2) for experiments, it would be nice to report flocs or any other time measurement for your method (not the inference) and competing methods.
3) Potentially you can apply your idea of pruning only one structure at a time (line 155 in Algorithm 1)  to any of the existing methods, so ablation is needed (is your improvement comes from the fact that u do essentially more gradual pruning?)
4) The method you propose does not seem to be LLM specific - why not to compare it with methods on structured pruning for non LLM models


- I am confused about contribution on line 56 (produces a family of compressed models). How is it any different from say using gradual pruning with various levels of sparsity (and at each step updating the copy for that sparsity level). You are still saving/updating multiple copies of the models right?


Minor:
- Algorithm 1: Mask Mr is not defined and is not updated in the body of the algorithm. I think you want to compile it based on R at the end of k iterations (the same also applies to Ms - please define how this mask looks like and its shape)

---

> ### Author Rebuttal · Authors · 2023-08-09
>
> **Question 1: Computational Cost**
>
> Please note that in the section “5 Discussion and Extensions”, paragraph “Computational efficiency”, we report two additional metrics: end to end runtime and efficiency with respect to the number of epochs, comparing also against the strongest competitor, CoFi:
>
> - In terms of the total number of epochs, ZipLM is more efficient than CoFi by a factor of _4.87x_ on larger tasks and _14.5x_ on smaller tasks.
> - In terms of the end to end runtime, ZipLM is very efficient as well: for large GLUE tasks, it produces all 14 compressed models in only 35 hours on a single GPU, while for the small tasks it does so in only 10 hours.
>
> **Question 2: Ablations**
>
> In our submission, “Appendix B: Ablation Studies” presents results of an ablation study on the gains from our distillation technique and demonstrates improvements up to _2 points_ in accuracy. Additionally, in Figure 2 of the response PDF, we focus on ablations specifically for the pruning metric. We compare results when pruning for sparsity (like prior approaches did) and when pruning for speedup (the ZipLM approach). The results suggest that the choice of pruning for speedup brings significant improvements, up to _10 points_, especially at higher speedups where inference-awareness is very important.
>
> **Question 3: How does pruning two consecutive layers happen?**
>
> In step 1 of our framework (Section 3.1), we indeed consider individual layers in isolation; but account for all correlations between different channels within each to perform accurate local pruning. However, in step 2 (Section 3.2) we also handle global interactions between layers by constructing (in an inference-aware manner) and evaluating many global pruning candidates based on the layer-wise solutions produced in step 1. Thus, overall, ZipLM captures both local as well as global correlations, which is a key feature of our algorithm.
>
> **Question 4: Formulation (1) vs. original OBS**
>
> Yes, this is correct. In the local pruning step (Section 3.1), we apply similar techniques to the OBS, but in the context of the layer-wise pruning problem defined in equation (1). For this particular formulation, the Hessian, which OBS approximates via Fisher gradients, can be calculated directly as XX^T (and our OBS is thus exact in this case). We will improve the clarity of this aspect in the next revision.
>
> **Question 5: Potentially you can apply your idea of pruning only one structure at a time to any of the existing methods.**
>
> The idea of pruning only one structure at a time cannot be applied to most other structured pruning approaches because they do not apply weight updates to remaining weights during the pruning step (e.g., they simply remove structures with lowest saliency); hence there would be no benefit of such a strategy. In contrast, in ZipLM, after removing the first structure, we can very efficiently update the remaining weights to compensate for this removal and re-evaluate their scores before the next selection. We would further like to emphasize that this one-at-a-time removal is only enabled by the high efficiency of our algorithm; applying this idea to other approaches which compute updates in less efficient ways would be far too slow (e.g., the MLP-layer of BERT-base has > 3k structures to remove one-at-a-time).
>
> **Question 6: The method you propose does not seem to be LLM specific - why not to compare it with methods on structured pruning for non LLM models**
>
> It is true that aspects of our method are general, and could be extended to other model types (CNNs or ViTs). We have chosen to focus on LLMs in our practical comparison since they are a clear focus for efficiency research these days, and, consequently, we have several high-quality baselines to compete against.
>
> **Question 7: I am confused about contribution on line 56 (produces a family of compressed models).**
>
> This contribution warrants special attention because all other structured pruning methods necessitate _a complete gradual pruning process for each sparsity target independently_, since intermediate checkpoints are not accurate enough. In contrast, ZipLM achieves the generation of all models within a single run, producing remarkably accurate intermediate models featuring varying levels of sparsity.
>
> To elaborate, if one were to obtain models with compression ratios of 2x, 3x, ..., up to 15x using the CoFi method, it would entail distinct runs for each ratio: one for 2x, another for 3x, and so forth, up to 15x.
>
> **Question 8: Please define how the masks look like and their shapes**
>
> We use $M_A$ to denote a pruning mask where all entries corresponding to the set of indices in $A$ are 1 and other elements 0; e.g., $M_R$ is a mask of all remaining indices after pruning. We will make this more clear in the next revision.
>
> **Question 9: Not very clear novelty**
>
> Regarding novelty, we would like to note a few additional points, relating to the answers to your questions above:
>
> - From the practical perspective, ZipLM is the first _gradual structured pruning approach_ which enables it to produce the entire Pareto frontier of models a lot more efficiently than the previous state-of-the-art method CoFi. As we detailed in Question 1 and Question 7, ZipLM is _4.87x_ more efficient on larger datasets and _14.5x_ more efficient on smaller datasets.
> - This advancement over prior work is enabled by a number of conceptual innovations: a new highly accurate pruner (as shown to be state-of-the-art in one-shot scenario), implemented by a highly efficient algorithm (as demonstrated by the ability to do one removal at a time with re-evaluation of the pruning scores), and a new distillation technique. All of this is complemented by inference-awareness (as demonstrated in the ablation study in Figure 2 of the response PDF).

---

> > ### Author Response · Authors · 2023-08-21
> >
> > Dear Reviewer,
> >
> > Given that we did not get the opportunity to interact during the discussion period, we briefly summarize our response:
> >
> > 1. Our rebuttal provides data and experimental results which address your concerns. Specifically, our algorithm is anywhere from ~5x to ~14x more efficient than previous state-of-the-art approaches, while at the same time providing superior results across the board.
> >
> > 2. This ability is unlocked by a number of new novelty aspects relative to prior work: specifically, a highly accurate and highly efficient structured pruner, complemented with inference-awareness step for the best accuracy-speedup tradeoff.
> >
> > We sincerely hope that you will acknowledge our response and additional results.
> >
> > Thank you for your time and effort, \
> > The ZipLM authors.

---

> > ### Comment · Reviewer_rTQd · 2023-08-21
> >
> > Thank you for your response.
> > Do you have any references to backup this statement "necessitate a complete gradual pruning process for each sparsity target independently, since intermediate checkpoints are not accurate enough. " or may be experimental results? E.g. any results for the intermediate checkpoints from other methods vs your intermediate checkpoints that you claim are much better quality?

---

> > > ### Author Response · Authors · 2023-08-21
> > >
> > > Dear Reviewer,
> > >
> > > Yes, we can certainly support this statement.
> > >
> > > Please examine the GitHub repository of CoFi, the prior state-of-the-art approach: https://github.com/princeton-nlp/CoFiPruning#:~:text=An%20example%20for,script%20for%20evaluation .
> > >
> > > It is clear that, to produce one sparse model, one has to run the entire pruning+finetuning pipeline.
> > >
> > > Moreover, this is directly confirmed by the main author of the paper, in the following comment: https://github.com/princeton-nlp/CoFiPruning/issues/2#:~:text=Hi%2C,a%20specific%20sparsity .
> > >
> > > Specifically, the text:
> > > > "CoFi requires training a single model every time for a specific sparsity"
> > >
> > > By contrast, with ZipLM we produce all models in a single run. Specifically, the ZipLM results illustrated across all figures are results of intermediate checkpoints, whereas the results with other approaches (like CoFi or any other distillation-based method) are obtained via one full run for each sparsity target. To our knowledge, ZipLM is the only method which produces all checkpoints in a single run.

---

> > > > ### Comment · Reviewer_rTQd · 2023-08-21
> > > >
> > > > Thanks, I appreciate your quick response and therefore raising my score

---

### Official Review · Reviewer_h7Df · 2023-07-06

**Soundness:** 4 excellent
**Presentation:** 2 fair
**Contribution:** 3 good
**Rating:** 6
**Confidence:** 3

**Summary:**

The paper proposes ZipLM, a structured pruning method that can achieve desired target runtime speedups.​ The idea of ZipLM is to solve a layer reconstruction problem with a structural constraint which minimizes the output changes on a set of calibration examples if the layer is reconstructed as it. The proposed ZipLM is shown to be effective on both decoder-only models and encoder-only models, outperforming prior distillation approaches and structured pruning approaches.

**Strengths:**

* The paper focuses on structured pruning problems, where the pruned model can easily get real speedups. Moreover, the proposed approach takes as input a target speedup as well as the hardware environment and optimizes the model on this specific setup to ensure the model to achieve desirable speedup. I admire this realistic setting and believe that this can be practically impactful.
* The paper has conducted extensive experiments studying the effectiveness of the proposed ZipLM approach, including on both encoder and decoder models, both one-shot pruning setting and gradual compression setting, and with different levels of sparsity.
* The experimental results show the proposed approach is more effective compared to the existing baseline methods.


**Weaknesses:**

* I believe the paper (especially the methodology part) can be presented better and a lot more details should be added (even in the appendix). For example, how do you solve the reconstruction problem and obtain the optimal mask and weight update (equation (2) & (3))? How do you consider the constraint C in equation (1) in the solution equation (2) & (3)? What exactly does a latency table look like and how do you get the table given a hardware environment? How many calibration inputs did you use in your experiments and how sensitive the final results will be to the number/quality of the calibration inputs?
* The paper is motivated by structured compression of large language models (LLMs). However, all experiments are conducted on models with hundreds of millions of parameters. Given the current state-of-the-art language models (e.g., LLAMA-7B/13B/65B) contain a couple orders of magnitude more parameters compared to BERT, it is not clear how the proposed method works in larger models.


**Questions:**

* Do you think the proposed approach can apply to larger language models such as LLAMA-7B? If not, what is the barrier?
* How sensitive the proposed approach is to the calibration examples?


**Limitations:**

I didn’t see a clearly potential negative societal impact of this paper.

---

> ### Author Rebuttal · Authors · 2023-08-09
>
> **Question 1: I believe the paper (especially the methodology part) can be presented better and a lot more details should be added (even in the appendix). For example, how do you solve the reconstruction problem and obtain the optimal mask and weight update (equation (2) & (3))? How do you consider the constraint C in equation (1) in the solution equation (2) & (3)?**
>
> Thank you for the comments on the presentation, which we will follow in the next revision.
> The main idea for deriving (2) and (3) is that (1) consists of the sum over row-wise linear regression problems. For linear regression, the exact error incurred by removing a set of weights as well as the optimal adjustment to the remaining weights can be determined in closed form, e.g., via solving a Lagrangian. Then, to couple structures across rows, we utilize the fact that there is no interaction between rows in the error of (1). Consequently, we can sum up the per-row formulas in appropriate ways to introduce our required coupling.
>
> (2) and (3) provide optimal formulas for removing a *single* structure; we then efficiently iterate those formulas until we have reached the desired sparsity target dictated by $C$ (we note that this iteration is not necessarily globally optimal but appears to be a good approximation in practice).
>
> In the next revision, we will improve the presentation of this part and include more details in the Appendix.
>
> **Question 2: What exactly does a latency table look like and how do you get the table given a hardware environment?**
>
> For a given model, batch-size, sequence-length and a device, we measure time of the forward pass through the main components of the Transformer layer, running in isolation: the attention and the FFN module. For the attention module we measure timings for varying number of heads, and for FFN module for varying intermediate size.
> The supplementary material of our submission contains an example of such a latency table at: code/bertbase_squad_V100.txt. Since the table is formatted such that it can be directly consumed by ZipLM algorithm, we provide a human readable example here:
>
> | Intermediate size | Latency (ms) |     $\|$      | Num of heads | Latency (ms) |
> | :-----------------: | :------------: |--- | :------------: | :------------: |
> | 3072              | 11.9         | $\|$ | 12           | 7.9          |
> | 1814              | 7.4          | $\|$ | 10           | 6.7          |
> | 1322              | 5.8          | $\|$ | 8            | 5.8          |
> | 302               | 1.6          | $\|$ | 6            | 4.4          |
> | 130               | 1.0            | $\|$ | 4            | 3.2          |
> | 76                | 0.9          | $\|$ | 2            | 1.9          |
> | 33                | 0.7          | $\|$ | 0            | 0            |
>
> **Question 3: Do you think the proposed approach can apply to larger language models such as LLAMA-7B? If not, what is the barrier?**
>
> In principle, there is no major obstacle to appling ZipLM to massive models such as LLAMA-7B, and in fact we believe this is a very interesting question for future work. Our results show that the method translates to good speedups both for generative inference (ZipGPT2) and for large-batch scenarios (ZipBERT). The main challenge to porting ZipLM to billion-scale models would be computational: since structured compression methods require significant fine-tuning for best results, one would need to have the computational budget (as well as sufficient GPU memory to train with optimal settings) to reproduce a fraction (e.g. 5%) of the original training in order to allow for good recovery.
>
> **Question 4: How sensitive the proposed approach is to the calibration examples?**
>
> We have found the method to be very robust to the amount of calibration data.
> To illustrate this, in the table below, we present a sensitivity analysis with respect to the number of calibration samples. We one-shot prune the fine-tuned BERT-base model on the SQuADv1 task for two speedup targets: 1.5x and 2.0x. In this setup, we compare results with the current state-of-the-art one-shot pruning approach of Kwon et al, which uses 2048 samples by default. As can be seen from the table, ZipLM outperforms prior state-of-the-art starting at only 32 samples. As we increase the number of samples, the results improve, up to 2 points in F1 score.
>
> |       Speedup = 1.5x    |   |  $\|$  |  Speedup = 2.0x | |
> | :----------------: | :-------- | :--------------: | :--------------: | :-------------: |
> | Num samples      | F1 score |     $\|$           | Num samples | F1 score |
> | ZipLM, 4         | 82.33    |        $\|$        | ZipLM, 4 | 48.41 |
> | ZipLM, 32        | 86.76    |        $\|$        | ZipLM, 32 | 82.63 |
> | ZipLM, 128       | 86.79    |      $\|$          | ZipLM, 128 | 83.56 |
> | ZipLM, 512       | 86.79    |       $\|$         | ZipLM, 512 | 84.06 |
> | ZipLM, 2048      | 87.08    |       $\|$         | ZipLM, 2048 | 84.14 |
> | ZipLM, 4096      | 87.62    |      $\|$          | ZipLM, 4096 | 84.68 |
> | Kwon et al, 2048 | 86.20     |     $\|$           | Kwon et al, 2048 | 76.50 |

---

### Official Review · Reviewer_mFy7 · 2023-07-07

**Soundness:** 3 good
**Presentation:** 3 good
**Contribution:** 3 good
**Rating:** 5
**Confidence:** 4

**Summary:**

This paper proposes ZipLM, a structured pruning and reconstructing + layer-wise distillation + inference-aware pruning algorithm. The authors first extend the pruning formula of OBS to structured pruning and utilize estimation of inference for each structure and structured SPDY search to achieve more accurate inference-aware pruning. Then, the authors distill the model in a token-wise manner. Experimental results demonstrate that ZipLM achieves state-of-the-art (SOTA) performance in both one-shot and pruning/knowledge distillation settings.

**Strengths:**

1. ZipLM demonstrates exceptional results among existing compression algorithms. It not only outperforms retraining-constrained approaches but also surpasses the performance of some pruning + knowledge distillation algorithms.
2. The authors propose a systematic framework for structured pruning, encompassing aspects such as pruning and reconstruction, inference-aware structure search, and knowledge distillation for performance recovery. This comprehensive framework exhibits a well-designed structure that, in my opinion, contributes to the entire community.

**Weaknesses:**

1. The methodology is somewhat incremental. The core contributions of the authors revolve around extending previous methods (OBS, SPDY) to structured pruning since those methods couldn't be directly applied. While the authors have made these extensions, the core essence of the method remains largely based on the previous framework.
2. The experimental section of this paper has some shortcomings. For instance, although the authors conducted a simple ablation experiment, they did not analyze which specific parts of the framework played a crucial role in improving performance. Additionally, in the majority of compression papers, the GLUE benchmark's eight datasets are evaluated as a whole since they represent the fundamental measure for assessing the performance of compressed models. However, the authors only evaluated four of these datasets.

**Questions:**

1. Could you provide a more comprehensive ablation experiment, specifically to identify which module within the entire framework contributes significantly to performance improvement?
2. Since Kwon [1] also employed a pruning metric and the layer-wise reconstruction via LLS for structured pruning, have you directly compared your pruning metric and the performance after reconstruction of OBS with their method to determine which one performs better, apart from the effect of distillation?
3. Line 192-193: For example, a 95% sparse BERT produced by CoFi has 12x speedup on a V100 GPU, but only 5x on an A100 GPU. Since this is an interesting observation that FLOPs-based calculation != inference speed, can you explain what is the potential reason of this?

---

> ### Author Rebuttal · Authors · 2023-08-09
>
> **Question 1: Ablations**
>
> In our submission, “Appendix B: Ablation Studies” presents results of an ablation study on the impact of distillation during the fine-tuning stage. As can be seen, our distillation technique can improve results up to _2 points_ in accuracy. Additionally, in Figure 2 of the response PDF, we focus on ablations specifically for the pruning metric. We compare results with ZipLM pruning when the target for pruning is sparsity (like prior approaches did) and when the target for pruning is speedup (the ZipLM approach). As can be seen from the Figure, the choice of pruning for speedup brings significant improvements, up to _10 points_, especially at higher speedups where inference-awareness is very important. For additional ablations on inference-awareness, please see our answer below for the difference between V100 and A100 speedups.
>
> In summary, these ablation studies decouple improvements from the three major components of our framework: token distillation, pruning metric, and inference-awareness. Please let us know if you would like to see any additional ablation studies, and we will be more than happy to provide them during the discussion period.
>
> **Question 2: Evaluation on the remaining tasks in the GLUE suite**
>
> We ran ZipLM on the remaining four tasks from the GLUE benchmark suite (CoLA, MRPC, STS-B, and RTE) and presented results in Figure 1 of the response PDF. As can be seen, ZipLM provides state-of-the-art results across these datasets as well, especially at higher speedup targets relative to prior structured pruning and distillation based approaches.
>
> **Question 3: Comparison of pruning metrics with Kwon et al. without distillation**
>
> We would like to highlight that we already show in “Table 2: One-shot structured pruning of BERT-base” that ZipLM outperforms the prior state-of-the-art one-shot approach of Kwon et al., in a setup without fine-tuning and thus also without distillation. In this setup only pruning metric and reconstruction are being compared.
>
> **Question 4: Difference in speedups on V100 vs A100**
>
> This discrepancy for CoFi models arises because the A100 GPU is significantly more powerful and thus faster on the dense model; at the same time, it is highly underutilized for small matrices, which significantly limits the speedups for very high sparsity. To illustrate this, we have measured the speedup from reducing the MLP size for both GPU types (see Table below). As can be seen, pruning to ~90% sparsity (3072 -> 302) gives ~7x speedup on a V100 but only ~3x speedup on an A100.
>
> *Table 1: Speedup improvements from shrinking the intermediate size of MLPs in the FFN section of a Transformer layer.*
> | MLP size | V100 | A100 |
> | :--------: | :----: | :----: |
> | 3072     | 1.0    | 1.0    |
> | 1814     | 1.6  | 1.1  |
> | 1322     | 2.0   | 1.4  |
> | 302      | 6.9  | 3.1  |
> | 130      | 11.8 | 4.4  |
> | 76       | 13.1 | 4.4  |
> | 33       | 14.8 | 4.4  |
>
> Such discrepancies are captured by inference-awareness of ZipLM approach, where pruning for sparsity is replaced by pruning for speedup which can utilize this information to guide pruning decisions.
>
> **Question 5: Novelty**
>
> Regarding novelty, we would like to note a few additional points, relating to the answers to your questions above:
>
> - Conceptually, ZipLM introduces two innovations relative to prior instances of the Optimal Brain Surgeon (OBS) framework: it focuses on structured pruning (so it can produce speedups on any hardware), and it is inference-aware (so it directly relates accuracy loss with real-world speedup gains). This is enabled by new technical derivations for the layer-wise structured compression problem, and by algorithmic insights which serve to speed up the combinatorial search (see the discussion in lines 172-182).
> - In addition, we also propose a very effective form of distillation for structured pruning.
> Practically, this leads to a very powerful framework, as we can produce the entire Pareto frontier corresponding to accuracy-vs-compression a lot more efficiently than prior methods: as shown, our method is on average _10 times more computationally efficient_ for this task than CoFi, the prior SOTA method.
> - As illustrated by the V100/A100 runtime discrepancy, a key differentiating feature of ZipLM is taking inference characteristics directly into account during compression. As such, ZipLM is the first method that yields state-of-the-art results for all 8 GLUE tasks, question answering (SQuAD), and text generation (WikiText), across BERT-base, BERT-large and GPT2 models, for both GPU and CPU deployment targets.

---

### Official Review · Reviewer_27Y8 · 2023-07-07

**Soundness:** 3 good
**Presentation:** 2 fair
**Contribution:** 2 fair
**Rating:** 6
**Confidence:** 3

**Summary:**

This paper proposes a structured compression method to optimize inference efficiency for language models. The proposed method uses the accuracy-efficiency trade-off under specific inference objectives as the importance measure of the model component( attention head, MLP neurons, entire FC/attention blocks.) Extensive evaluation results are provided to demonstrate the effectiveness of the proposed method.

**Strengths:**

I appreciate the thoroughness of the evaluation, in which the authors consider the encoder/decoder and decoder-only model,  retraining compression and zero-shot compression, throughput and latency objective, etc. Evaluation provides a better understanding of the proposed method.

**Weaknesses:**

This paper focuses on improving the language model's inference efficiency. Other than distillation and pruning, quantization is also a popular direction[1], which is not discussed/compared. Quantization generally reduces the model size. At the same time, a reduction in model size would require less number of GPUs for large model inference, leading to latency speed up.  Further, there is a line of work on dynamic sparsity[2] that improves inference latency, which is also not discussed/compared.

[1]Frantar, Elias, et al. "Gptq: Accurate post-training quantization for generative pre-trained transformers." arXiv preprint arXiv:2210.17323 (2022).
[2]Liu, Zichang, et al. "Deja vu: Contextual sparsity for efficient LLMs at inference time." (2023).

**Questions:**

Table 1 shows a significant drop in PPL compared to the original GPT2, even at the smallest speed-up. To better understand the trade-off, can the authors comment on at what speedup we can expect no loss in performance?

**Limitations:**

The authors didn't discuss limiation or negative social impact.

---

> ### Author Rebuttal · Authors · 2023-08-09
>
> **Question 1: Discussion of alternative approaches (quantization, Deja Vu)**
>
> Please note that quantization is complementary to both distillation and pruning; in practice, it is applied in conjunction with these two. Thus, rather than comparing pruned and quantized models, one can _combine_ these compression techniques to obtain compound improvements.
>
> This was our approach in this submission, where _we did apply quantization_ to ZipLM models. Specifically, in the “CPU as an LLM-inference environment” paragraph in section “5 Discussion and Extensions”, we describe how we combine ZipLM pruning with quantization. We used quantization-aware training (QAT) which, instead of quantizing only weights (like GPTQ), quantizes both weights and activations, thus enabling direct deployment of LLMs on edge devices as well as computational speedups for arbitrary batch sizes and number of input tokens (GPTQ only leads to speedups for generative inference at very low batchsize).
>
> Deja Vu leverages dynamic forms of sparsity, which appear at runtime, such as activation sparsity. Thus, Deja Vu is also complementary to ZipLM, which induces and leverages static structured sparsity. Further, while promising, Deja Vu comes with a few key limitations:
>
> (1) As of yet, DejaVu has only been shown to work on a few very specific models (OPT-30B/66B/175B and BLOOM-175B), which are extremely large and not very efficient to start with.
>
> (2) DejaVu requires complex custom CUDA kernels to produce runtime speedups, which are optimized for one-token-at-a-time inference, while speedups quickly diminish for larger batch-sizes or non-generative applications.
>
> The structured pruning approach we adopt is much more general; we have shown it to be applicable to both batch-prediction (throughput-constrained) and text-generation (latency-constrained) use cases. In addition to text-generation (GPT) use cases, our work also shows viability in various (non-generative) downstream tasks such as: text-classification, question-answering, spam detection, etc, for which Deja Vu currently _does not yield speedups_.
>
> In summary, both quantization and Deja Vu (contextual sparsity) are complementary to ZipLM. We have already shown compatibility with quantization in our submission (Section 5), and plan to investigate compatibility with contextual sparsity in future work: in particular, we plan to investigate whether ZipLM structured sparsity can induce significant additional contextual sparsity.
> We will add a clarifying discussion on both approaches in the next version.
>
>
> On a technical note: the Deja Vu paper was not publicly available at the time of the NeurIPS submission deadline, and therefore we would not have been able to cite or discuss it.
>
> **Question 2: PPL difference relative to the original GPT2**
>
> GPT2 has been trained by OpenAI on a much larger (> 10x) closed-source dataset, for significantly longer than both the DistilGPT2 and ZipGPT2. In contrast to that, DistilGPT2 and our ZipGPT2 models are trained on the same open-source dataset, for a much shorter period of time. Therefore the only possible direct comparison in this setup is between DistilGPT2 and ZipGPT2, which is clearly in favor of our approach: relative to DistilGPT2, we provide higher accuracy at the same speedup, and higher speedup for the same accuracy.
>
> In fact, to support these claims we pretrain GPT2 model from scratch on the open-source dataset used by DistilGPT2 and ZipLM. After evaluating the resulting model in the same setup as the other two models, we are observing the perplexity of 38.5 (the closed-source trained GPT2 has 28.5). This means that when trained on the same data and under similar computational budget, DistilGPT2 and ZipLM models do exhibit competitive performance to the uncompressed GPT2 model.
>
> **Question 3: Limitations and social impact**
>
> Please note that, following the NeurIPS23 submission guidelines, we have provided a discussion of limitations and impact in the 'Appendix H: Broader Impact and Limitations'.

---

> > ### Comment · Reviewer_27Y8 · 2023-08-14
> > **Rebuttal Response**
> >
> > Thanks for the pointer to the quantization experiment and the additional perplexity of GPT2 trained under the same setup.
> >
> > A discussion of quantization LLM would be nice to include in Related Work, aside from distillation and pruning.

---

> > > ### Author Response · Authors · 2023-08-15
> > >
> > > Thank you for the useful suggestion, we will include a discussion of quantization as well as contextual sparsity (Deja Vu) in our Related Work section in the next version of the paper.
> > >
> > > Please let us know whether you have any additional concerns or questions that we can address during the discussion period.

---

### Author Rebuttal · Authors · 2023-08-09

We wish to sincerely thank the reviewers and the AC for their work, and for the valuable feedback. We have provided individual responses for each reviewer question. We outline answers to some common topics below:

- Regarding the discussion of limitations and potential negative social impact in our work (Reviewers **27Y8** and **5zkW**), we would like to clarify that these aspects were taken into account in our original submission. Following the NeurIPS 2023 Call for Papers, we have addressed them in a dedicated section called '*Appendix H: Broader Impact and Limitations*,' where we delve specifically into the limitations and potential negative social impact of our research.
- Similarly, the ablation studies on various components of our method were already present in ‘*Appendix B: Ablation Studies*’ which isolates the accuracy impact of e.g. distillation on the final accuracy of the model.

In addition, we have performed the following additional experiments, which further address the reviewer concerns:

- To address the question of Reviewer **mFy7**, we have run ZipLM on the remaining 4 GLUE tasks. The results are illustrated in Figure 1 of the PDF response, and show the same trends as the original tasks. Specifically, ZipLM provides significant gains over prior methods, especially at higher compression rates. Results also showcase the fact that our method is significantly more computationally-efficient than prior work: we were able to obtain the full trade-off for each task in a single execution, whereas prior work would need a separate execution for each target, requiring on average _10x more computation_.

- To address the questions regarding differences in speedup on different GPUs and using different metrics (Reviewers **mFy7** and **5zkW**) we have provided a detailed analysis of how these differences arise, and how pruning for size compares with pruning directly for speedup using our method (see also Figure 2 in PDF response). The results show that pruning directly for speed is critical to obtaining good practical performance at the same accuracy level. The full answers are provided in the individual responses.

- We also provide an ablation with respect to the number of samples, showing that our method is extremely stable with respect to sample complexity. Specifically, we can outperform the prior SOTA method of Kwon et al. (which uses 2048 samples by default) starting at just 32 samples.

We thank the reviewers again for their feedback, and look forward to the discussion.

---

### Author Response · Authors · 2023-08-20
**Discussion reminder**

Dear Reviewers,

Given that the discussion period is ending shortly, we wanted to send a gentle reminder regarding our review responses.

We would really appreciate your feedback on our rebuttal, especially regarding the added experimental results and additional clarifications regarding our method.

With best regards,

The ZipLM Authors

---

### Decision · Program_Chairs · 2023-09-21

**Decision:**

Accept (poster)

**Comment:**

This paper proposed a structural pruning method for language models, with the consideration of runtime at the inference time.
The reviewers generally like the comprehensive studies offered in this paper, and the good empirical results obtained. However the concerns are around the novelty and technical contribution w.r.t. the prior works. Nevertheless, this paper has made good contribution to the field of network pruning, and I encourage the authors to refine the paper further by incorporating the reviewers' comments into the revisions.